# Deglacial evolution of regional Antarctic climate and Southern Ocean conditions in transient climate simulations

Daniel P. Lowry[1], Nicholas R. Golledge[1,2], Laurie Menviel[3], Nancy A.N. Bertler[1,2]

[1]Antarctic Research Centre, Victoria University of Wellington, Wellington, 6012, New Zealand
[2]GNS Science, Lower Hutt, 5010, New Zealand
[3]Climate Change Research Centre and PANGEA Research Centre, University of New South Wales, New South Wales, 2052, Australia

*Correspondence to*: Daniel P. Lowry (Dan.Lowry@vuw.ac.nz)

**Abstract.** Constraining Antarctica's climate evolution since the end of the Last Glacial Maximum (~18 ka) remains a key challenge, but is important for accurately projecting future changes in Antarctic ice sheet mass balance. Here we perform spatial and temporal analysis of two transient deglacial climate simulations, one using a fully coupled GCM (TraCE-21ka) and one using an intermediate complexity model (LOVECLIM DG$_{ns}$), to determine regional differences in deglacial climate evolution and identify the main strengths and limitations of the models in terms of climate variables that impact ice sheet mass balance. The greatest continental surface warming is observed over the continental margins in both models, with strong correlations between surface albedo, sea ice coverage and surface air temperature along the coasts, as well as regions with the greatest decrease in ice surface elevation in TraCE-21ka. Accumulation-temperature scaling relationships are fairly linear and constant in the continental interior, but exhibit higher variability in the early to mid-Holocene over coastal regions. Circum-Antarctic coastal ocean temperatures at grounding line depths are highly sensitive to the meltwater forcings prescribed in each simulation, which are applied in different ways due to limited paleo-constraints. Meltwater forcing associated with the Meltwater Pulse 1A (MWP1A) event results in sub-surface warming that is most pronounced in the Amundsen and Bellingshausen Sea sector in both models. Although modelled centennial-scale rates of temperature and accumulation change are reasonable, clear model-proxy mismatches are observed with regard to the timing and duration of the Antarctic Cold Reversal (ACR) and Younger Dryas/early Holocene warming, which may suggest model bias in large-scale ocean circulation, biases in temperature reconstructions from proxy records, or that the MWP1A and 1B events are inadequately represented in these simulations. The incorporation of dynamic ice sheet models in future transient climate simulations could aid in improving meltwater forcing representation, and thus model-proxy agreement, through this time interval.

## 1. Introduction

Ice sheet model simulations of both past and future climates often rely on paleoclimate forcings derived from a combination of proxy data and climate model simulations (Pollard and DeConto, 2009; Golledge et al., 2014; Gregoire et al., 2016; Bakker et al., 2017). As such, the long memory of ice sheets with respect to past climatic conditions means that a model spin-up with an accurate deglacial climate forcing is important for any future Antarctic ice sheet model simulations. However, paleoclimate proxy data generally have sparse spatial coverage and relatively large uncertainties. Regional differences also persist in many of the Antarctic ice core records due to local climate effects (Pedro et al., 2011; Veres et al., 2013), complicating their use as climate

forcings for an entire ice sheet. In addition to the ice core record, the marine proxy record is limited in aiding ice sheet modellers due to the corrosive nature of Southern Ocean bottom waters, which inhibits the preservation of microfossils that can be dated to provide age controls or geochemically analysed to provide a proxy for ocean temperature (McKay et al., 2016). Although climate forcings derived from compilations of multiple records and global datasets can help alleviate some of these issues, the regional differences between proxy archives may actually be consequential to ice sheet mass changes (Golledge et al., 2017). For example, the Antarctic ice sheet retreat during the last deglaciation likely occurred asynchronously (The RAISED Consortium, 2014), suggesting that different sectors of the ice sheet responded either to different forcings, or in different ways to uniform forcings.

Climate model simulations can help address the data gaps in the proxy record, however, they are often assessed in terms of skill during climate states of relative stability, such as the Last Glacial Maximum (LGM), the mid-Holocene or the Last Millennium (Braconnot et al., 2012; Schmidt et al., 2012; Hargreaves et al., 2013; Sueyoshi et al., 2013), rather than over long-term periods of dramatic climate change. In fact, for the last deglaciation, only one series of transient climate simulations using a fully coupled atmosphere-ocean general circulation model (GCM) has been performed (Liu et al., 2009; He, 2011; He et al., 2013), and it may miss important synoptic-scale processes related to the surface mass balance of West Antarctica (Fudge et al., 2016). It also shows some discrepancies related to the timing and magnitude of the Antarctic Cold Reversal (ACR; He, 2011). Given that the task of simulating global climate with a variety of evolving internal and external forcings is computationally demanding, the use of intermediate complexity models is especially appealing, and such models have been successfully applied to better understand past climate changes (Menviel et al., 2011; Goosse et al., 2012; Menviel et al., 2014). However, the coarser resolution and relatively simpler parameterization schemes of the atmospheric models may introduce further challenges with regard to simulating the climatic processes that affect ice sheet mass balance.

In this study, we evaluate the output of two transient deglacial climate simulations, one using a fully coupled GCM (Liu et al., 2009) and one using an Earth system model of intermediate complexity (Menviel et al., 2011). Given that the mass balance of the West and East Antarctic ice sheets is largely controlled by the accumulation of snow on the surface minus the ice that is lost through sublimation, iceberg calving, sub-ice shelf melt and thermal erosion of marine-based grounded ice, we focus on the aspects of climate that are most relevant to these processes, including surface temperature, surface mass balance, ocean temperatures to grounding line depths and sea ice. This analysis is performed in the context of the Antarctic ice core and Southern Ocean marine sediment records. The main goals of this study are to (1) determine the regional differences in the deglacial climate evolution of Antarctica and the Southern Ocean as recorded in the models and proxy records, and (2) identify the main strengths and biases of the models in capturing the rates and magnitudes of climatic changes that impact ice sheet mass balance. We focus our analysis on the period from the last glacial termination (18 ka; Denton et al., 2010) to the mid-Holocene (6.5 ka).

## 2. Materials and Methods

### 2.1 Transient climate model simulations

In this study, we consider two transient deglacial climate simulations, namely, the TraCE-21ka and LOVECLIM DG$_{ns}$ deglacial experiments (Table 1). TraCE-21ka is a transient climate simulation of the last

22,000 years using the Community Climate System Model version 3 (CCSM3), a synchronously coupled GCM with atmosphere, ocean, land surface and sea ice components and a dynamic global vegetation module (Collins et al., 2006; Liu et al., 2009; He, 2011). The transient forcings included evolving orbital forcing following Milankovich theory (Genthon et al., 1987), greenhouse gas concentrations ($CO_2$, $CH_4$, $N_2O$; Joos and Spahni,

2008), ice sheet and paleogeography changes based on the ICE-5G reconstruction (Peltier, 2004), and prescribed meltwater fluxes into the Atlantic, North Atlantic and Southern Oceans (see He, 2011).

      The LOVECLIM $DG_{ns}$ experiment was performed with the intermediate complexity LOVECLIM model version 1.1 (Driesschaert et al. 2007; Goosse et al. 2007), which includes a quasi-geostrophic atmosphere model, an ocean general circulation model, a dynamic/thermodynamic sea ice model as well as ocean carbon

cycle and terrestrial vegetation components (Menviel et al. 2011). The horizontal resolution of the atmosphere component is coarser than that of TraCE-21ka (spectral resolution of T21 compared to T31), with 3 levels in the atmosphere, but those of the ocean and sea ice components are similar (Table 1). The transient forcings included time-varying solar insolation (Berger, 1978), Northern Hemisphere ice sheet topography updated every 100 years (Peltier, 1994), and atmospheric $CO_2$ determined from the EPICA Dome C ice core (Monnin et al. 2001),

with $CH_4$ and $N_2O$ fixed at LGM levels. Freshwater pulses were applied to the North Atlantic and Southern Ocean based on $^{231}Pa/^{230}Th$ data of the North Atlantic (McManus et al., 2004) and Greenland temperature reconstructions (Alley, 2000). Two transient simulations were performed; one with freshwater input in the Southern Ocean at the time of the ACR, and one without; we consider the former (see Menviel et al., 2011). We also note that the ice sheet topography over Antarctica was unchanged in this simulation. Additional model

details of TraCE-21ka and LOVECLIM are presented in Table 1.

      In terms of the transient forcings, one of the most uncertain aspects concerns the timing, magnitude and location of meltwater fluxes, which are handled in the two simulations in different ways. In both models, the freshwater is applied over large areas of the ocean surface in order to capture millennial-scale discharge events that occurred during the deglacial period, but the amounts and locations vary due to limited paleo-constraints. In

the early part of the study period (18-14.5 ka), a prescribed freshwater flux into the North Atlantic in TraCE-21ka is increased to ~0.17 Sv, at which point it is held constant until 14.67 ka. $DG_{ns}$ likewise has a prescribed freshwater flux into the North Atlantic during this time interval, however the flux ceases earlier than in TraCE-21ka (0.2 Sv from 18-17.4 ka; 0.25 Sv from 17.4 to 15.6 ka).  Meltwater Pulse 1A (MWP1A) is also represented differently: in TraCE-21ka, very high, short-lived freshwater fluxes into the Ross and Weddell Seas occur

between 14.35 and 13.85, peaking at ~14.1 ka (0.33 Sv each), with lower freshwater forcing applied to the Mackenzie River and Gulf of Mexico regions (reaching 0.11 Sv each). In contrast, in $DG_{ns}$ a higher freshwater flux is applied in the North Atlantic as compared to those in the Ross and Weddell Seas (0.2-0.25 Sv vs. 0.15 Sv, linearly decreasing to 0 Sv, respectively), and the meltwater forcing occurs over a longer duration than in TraCE-21ka (14.4 to 13.0 ka). For the Younger Dryas, an Arctic Ocean meltwater forcing is applied from 13-

12.2 ka in $DG_{ns}$ (0.25 Sv), whereas in TraCE-21ka, a lower meltwater forcing is applied in the mid-latitude St. Lawrence River region from 12.9 to 11.7 ka (0.17 Sv). In the Holocene, TraCE-21ka has a large meltwater forcing of 5 Sv in the Hudson Strait region at 8.47 ka for half a year to represent the 8.2 ka event, but no such forcing is applied in $DG_{ns}$. Additional details on the prescribed meltwater forcings in the $DG_{ns}$ and TraCE-21ka experiments can be found in Menviel et al. (2011) and He (2011), respectively.


### 2.2 Spatial and Temporal Analysis

Considering differences observed between Antarctic ice core records from East and West Antarctica and between coastal and interior regions, we focus our analysis on the following four continental regions (see Fig 1): the interior of the East Antarctic ice sheet (EAIS interior; 83°S-75°S, 30°W-165°E), coastal East Antarctica (coastal EAIS; 75°S-68°S, 15°W-165°E), West Antarctica (WAIS; 83°S-72°S, 165°E-30°W), and the Antarctic Peninsula (AP; 72°S-64°S, 64°W-59°W). We assess the magnitudes and rates of continental surface temperature and accumulation changes in the climate simulations from the start of the glacial termination (18 ka; Denton et al., 2010) to the mid-Holocene (6.5 ka), a period of time covered by both climate model simulations.

Here, we define accumulation as precipitation minus surface evaporation/sublimation (P – E). Accumulation from the ERA-Interim Analysis defined as precipitation minus sublimation has been shown to closely match airborne radar observations over the Thwaites Glacier in West Antarctica (Medley et al., 2013), however, we note that the contribution of wind-transported snow may also be locally significant in some areas, particularly in coastal areas dominated by katabatic winds (Palm et al., 2017). As such, we emphasize caution in interpreting accumulation changes in the climate models in areas of strong wind speed, but in terms of overall ice sheet mass balance, wind-transported snow is generally considered to be a relatively minor component (Palm et al., 2017). We also ignore the surface runoff term given that it is dependent on snow thickness thresholds that are set in the climate models, hence over long timescales, it balances P-E over Antarctica. For each continental region, we determine the scaling relationships between temperature and accumulation in both climate model simulations and compare these relationships to those recorded in the ice core records.

In addition to continental surface temperature and accumulation, we examine the coastal seas around Antarctica to depths corresponding to modern-day grounding lines (0-800 m), including the Ross Sea (70°S-62°S, 168°E-160°W), the Amundsen and Bellingshausen Seas (68°S-62°S, 135°W-60°W), the Weddell Sea (70°S-62°S, 60°W-30°W), the coastal region from Lazarev Sea to Cosmonauts Seas (67°S-62°S, 15°W-50°E), and the coastal region from Cooperation Sea to Somov Sea (67°S-62°S, 55°E-165°E), as well as the surface of the entire Southern Ocean. We identify changes in the surface ocean temperatures near the continental shelf as well as sea ice thickness, concentration and extent. Since the TraCE-21ka ocean and sea ice outputs are available at the decadal scale, we focus this analysis on decadal- to centennial-scale changes.

### 2.3 Proxy Records

To assess climate model performance, we rely on publicly available proxy records of Antarctic climate and the Southern Ocean conditions with reconstructions that are directly comparable to the climate model output (Table 2; Fig 1). Specifically, we use the temperature reconstructions of the Vostok (V; Lorius et al., 1995; Petit et al., 1999), Dome Fuji (DF; Uemura et al, 2012), and James Ross Island ice cores (JRI; Mulvaney et al., 2012), and the temperature and accumulation reconstructions of the EPICA Dome C (EDC; Jouzel et al., 2007; Parennin et al., 2007; Stenni et al., 2010) and West Antarctic Ice Sheet Divide ice cores (WDC; WAIS Divide Project Members, 2013; Cuffey et al., 2016; Fudge et al., 2016). Only ice core records with temperature and/or accumulation reconstructions that overlapped with the period covered in both models (18-6.5 ka) were used in this analysis. The exception is James Ross Island, which does not extend back to 18 ka, hence only the period of 14.2 to 6.5 ka is considered for this site.

The temperature reconstructions are commonly calculated as anomalies to modern temperatures based on deuterium ($\delta$D), deuterium excess ($d$) and oxygen isotope measurements ($\delta$O$^{18}$) using site-specific temperature-dependence relationships that assume the modern-day calibration holds over the entire record (Jouzel et al., 1997; Mulvaney et al., 2012). However, there is inherent uncertainty in this approach as the isotopic signals can be influenced by changes in moisture source regions (Uemera et al., 2012), sea ice content (Holloway et al., 2016), and the seasonality of precipitation (Erb et al., 2018). In East Antarctic ice cores, for example, temperature reconstructions have been estimated to have an uncertainty range of -10% to +30% (Jouzel et al., 2003), or +/-2°C for glacial-interglacial temperature change (Stenni et al., 2010). Recent improvements to this traditional method have been successfully applied to the WDC ice core, however, with the use of additional borehole temperature and nitrogen isotope data, yielding a temperature reconstruction with relatively lower uncertainty (+/-1.8 °C or 16%; Cuffey et al., 2016). The accumulation reconstructions are more difficult to obtain, but can be estimated from the depth-age relationship by correcting the annual-layer thicknesses for flow-induced thinning using an ice flow model (Parennin et al., 2007; Fudge et al., 2016). Sources of uncertainty in accumulation reconstructions include both the timescale and the amount of thinning, which can be quantified using a firn densification model (Fudge et al., 2016). Uncertainties of accumulation reconstructions for the deglacial period are estimated to range from ~15-25% (Freiler et al., 2015; Fudge et al., 2016).

In addition to Antarctic ice core records, we use marine sediment records from the Southern Ocean (Table 2; Fig 1). The records include alkenone-derived SST reconstructions from Core MD03-2611 off the coast of South Australia (Calvo et al., 2007), Core TN057-6 in the South Atlantic (Anderson et al. 2014), and Core MD07-3128 (Caniupan et al. 2011) and the Ocean Drilling Program (ODP) Core 1233 off the coast of Southern Chile (Kaiser et al., 2005), the Mg/Ca-derived SST reconstruction of Core MD97-2120 from Chatham Rise (Pahnke et al., 2003), and the diatom-derived seasonal SST reconstruction of Core TN057-13 from the East Atlantic Polar Front (Anderson et al. 2009). These SST reconstructions require cautious interpretations as they record temperatures at the depth at which the foramifera and diatoms live, and therefore may not offer straightforward reconstructions of SST. Other limitations to comparing these records to the model SST output include their relatively low temporal resolution, which can have millennial-scale gaps, uncertainties in their age scales, as well as the potential for seasonal biases. The alkenone-derived SSTs in particular can be biased to lower or higher values than the annual mean in regions of high seasonality in accordance with the seasonal cycles of marine archaea (Prahl et al., 2010). Existing calibrations have the tendency to overestimate SSTs at high latitudes, and even with recent improvements, uncertainties of alkenone-derived SSTs in the Southern Ocean over the Quaternary period are estimated to generally be >5°C (Tierney and Tingley, 2018). Comparisons between SSTs derived from different proxies, such as Mg/Ca, can also be challenging because they may be recording different seasonal signals (Leduc et al., 2010). Given that the true uncertainties of these marine records are difficult to quantify, we suggest caution with respect to their direct comparison to the climate model output.

**3. Results**

**3.1 Surface temperatures**

The DG$_{ns}$ and TraCE-21ka deglacial simulations both show fairly uniform temperature increases over the continent and Southern Ocean of >6°C (Fig. 2a,b). The exception is the continental interior in the DG$_{ns}$ simulation, which exhibits more modest temperature increases, particularly closer to the pole. In contrast, the TraCE-21ka simulation experiences the greatest warming over the pole and much of WAIS (>12°C). This difference is primarily driven by the change in ice sheet topography in TraCE-21ka; the decrease in surface elevation can explain most of the temperature increase, as indicated by a sensitivity simulation in which only the ice sheet topography is changed while all other boundary conditions remain the same (Supplementary Information). If the same topographic changes were applied to the DG$_{ns}$ simulation, ~2.5-6.5°C of the difference in temperature change between the two models in the continental interior could be accounted for, considering a glacial lapse rate of 0.54°C/100m calculated in DG$_{ns}$ over the continent. Both models simulate large temperature increases along the coasts over the Ross, Amundsen, Bellingshausen and Weddell Seas, as well as along parts of the East Antarctic coast (>10°C). DG$_{ns}$ displays a larger temperature increase over the Antarctic Peninsula and Weddell Sea (10-13°C), whereas TraCE21ka shows a larger temperature increase at Prydz Bay and the Ross Sea (~12°C).

In comparison to the temperature change estimated in the ice core records over the analysed period (6.5 – 18 ka), DG$_{ns}$ displays lower mean annual temperature changes at each site except for JRI (Fig 2a), whereas TraCE-21ka displays higher temperature changes at each site (Fig 2b). However, with the exception of V in DG$_{ns}$, the range of seasonal temperature change of both models (i.e., the change in austral summer and winter temperatures) is within the uncertainty of the temperature reconstructions at each site (Table 3). For the first 4 kyr of the simulation, DG$_{ns}$ has a warm bias in the EAIS interior in terms of the mean annual temperature anomaly close to the pole due to its use of a modern Antarctic ice sheet configuration and model edge effects (Fig 2c). In contrast, TraCE-21ka has a cold bias for the full extent of the analyzed time period for WAIS and until ~9 kyr in the EAIS interior (Fig 2c,e). Compared to the WDC ice core, TraCE-21ka underestimates the surface temperature change until a substantial increase occurs at 11.3 ka due to an abrupt decrease in the ice surface elevation. This occurs at a time of stable temperature in the ice core record (Fig 2e). Both models show cold biases for the AP, but DG$_{ns}$ converges with the JRI reconstruction in the early Holocene. Overall, the temperature anomalies are more similar between the model simulations and ice cores in the Holocene than earlier in the deglacial period.

In each region, the main discrepancy of the models with the Antarctic ice core temperature records is the timing and magnitude of the ACR. The TraCE-21ka simulation displays a sharp ACR following a large pulse of freshwater into the Southern Ocean and the Gulf of Mexico, with a decrease in surface temperature of ~2.5°C in continental Antarctica, and a minimum occurring at approximately 14 ka. The duration of the event is short-lived, lasting less than 500 years, as AMOC strength decreases back to its pre-meltwater forcing level (He, 2011). The ACR duration is longer in the DG$_{ns}$ simulation (~1-1.5 kyr), initiating at approximately the same time as in the TraCE21ka simulation, but the magnitude of the surface temperature change is lower, especially over the EAIS interior region (Fig 1c). In contrast to the simulations, the EAIS ice core records show a later initiation of the ACR, beginning at about 14 ka, rather than peaking at this time. The WDC ice core exhibits a similar initiation timing to the climate model simulations, but shows a more gradual decrease and subsequent increase in temperature (Fig 1e). The overall surface temperature change of the ACR is moderate in the ice core records as compared to the low (high) temperature change in DG$_{ns}$ (TraCE21ka), and the event lasts for

approximately 2 kyr, depending on the site. The JRI ice core shows the subtlest expression of the ACR, with the WDC, EDC and V ice cores displaying more pronounced signals. Both climate simulations show larger temperature changes associated with the ACR over the AP, which may be due to the strong influence of sea ice changes on surface temperature.

### 3.2 Surface mass balance

Precipitation increases in every grid cell over the continent and Southern Ocean in the TraCE-21ka simulation over the period from 18 to 6.5 ka, and these changes are significant at the 95% confidence level (Fig. 3a). The greatest increases occur over WAIS, coastal EAIS, and most of the Southern Ocean (>8 cm/yr). The AP, EAIS interior, and the Roosevelt Island region show modest increases (<4 cm/yr). The $DG_{ns}$ simulation shows a similar precipitation increase over the Southern Ocean (>8 cm/yr), however, precipitation decreases of 1-6 cm/yr occur over the South Pole and the coastal EAIS (Fig 3b). In the coastal region, where the decrease is larger, this is related to the coarse model resolution, which cannot adequately reproduce the steep slopes of East Antarctica and thus underestimates snow deposition in this region (Goosse et al., 2012). Changes in atmospheric circulation lead to the slightly reduced precipitation over the South Pole. Also, in contrast to TraCE-21ka, $DG_{ns}$ precipitation over the AP increases by >12 cm/yr, and increases by 4-10 cm/yr over the EAIS interior.

Modelled P-E anomalies relative to the preindustrial era (PI) are quite distinct in each region and differ to the accumulation reconstructions of the ice core records (Fig 3c-f). In general, the E term is substantially lower than the P term, hence E has a negligible effect on the surface mass budget. At 18 ka, both models generally show negative mean annual P-E anomalies relative to PI. The exception is the coastal EAIS region in the $DG_{ns}$ simulation, which shows higher relative mean annual P-E and a negative trend through time (Fig 3d). The models show the highest P-E variability over the AP, and show a pronounced decrease in mean annual P-E associated with the ACR (Fig 3f). In this region, TraCE-21ka remains near PI P-E levels, whereas $DG_{ns}$ shows a more substantial increase through time.

The overall accumulation change reconstructed at the WDC site for the analyzed period is similar to that of TraCE-21ka (Fig 3b,e), though the model has a negative P-E bias until the large ice sheet configuration change (decrease in surface elevation) that occurs at 11.3 ka. In contrast, $DG_{ns}$ underestimates the overall accumulation change at this site (3a,e). Accumulation recorded in the EDC ice core is relatively stable. In fact, Cavitte et al. (2017) highlights the stability of accumulation patterns at Dome C over the last glacial cycle, and the greatest change in the centennial-averaged record of annual average accumulation is only 2.17 cm/yr. TraCE-21ka shows a negative P-E anomaly relative to PI through this time interval, however, it exhibits a similar magnitude of change and variability in the deglacial period to the EDC accumulation record over the broader EAIS interior region (Fig 3c). In contrast to TraCE-21ka, the $DG_{ns}$ simulation overestimates the magnitude and variability of P-E in the EAIS interior.

Another discrepancy between the two model simulations is the larger difference between austral summer and winter accumulation anomalies in $DG_{ns}$ as compared to TraCE-21ka (color shading in Fig 3c-f). While this is true in each region, the most extreme case is the EAIS coastal region, in which the austral summer accumulation anomaly relative to PI is as high 40.8 cm yr$^{-1}$ at the start of the simulation, whereas the austral winter accumulation anomaly is -12.65 cm yr$^{-1}$. Through time, austral summer (winter) accumulation decreases

(increases) in this region, thereby reducing the seasonal anomaly range in the Holocene relative to the early deglacial period.

**3.3 Accumulation-temperature relationships and rates of change**

Centennial-scale rates of surface temperature and accumulation changes in the ice core records are well-matched by the climate model simulations, although the timing of the largest changes are generally offset (Fig 4a-d). This is particularly true for surface temperature, with the ice core records showing warming (cooling) not exceeding 0.4°C (-0.2°C)/100 yrs and the climate models showing a slightly larger range (-0.5 to 0.6°C/100 years in the case of TraCE-21ka). The greatest discrepancies between the ice cores and the models are the timing and magnitude of cooling and warming associated with the ACR and an artificial warming spike in TraCE-21ka associated with the change in ice surface topography at 11.3 ka (Fig 4a-d). The $DG_{ns}$ simulation also shows a high warming rate following the ACR in each region, however, this is not due to any change in ice surface elevation. The models show similar variability in the precipitation changes (% relative to PI)/100yrs to those of the ice core accumulation records (Fig 3e-h). Of note is the increase and subsequent decrease in the accumulation rate that occurs following the ACR, which is present in both the WDC and EDC records, which slightly exceed the climate model precipitation rate variability (Fig 4e,g). As expected, the models both show substantially higher variability in the rate of precipitation change along the coasts.

In terms of the relationship between temperature and accumulation, Frieler et al., (2015) demonstrated that in 5 East Antarctic ice core records of the deglacial period, a consistent positive linear scaling relationship exists, with an average of ~6.0% °C$^{-1}$, in accordance with that expected of the Clausius Clapeyron relationship. The TraCE-21ka simulation captures these relationships at these individual sites relatively well, generally falling within the range of uncertainty in the ice core record. Here, we show that the $DG_{ns}$ simulation using the intermediate complexity model can likewise reproduce this positive scaling relationship in the EAIS interior, however, the model fails in the coastal region of the EAIS (Fig 5a,b). This is explained by the model resolution issue mentioned previously, which is enhanced through time as the moisture gradient between the EAIS interior and the Southern Ocean is reduced.

In contrast to the EAIS ice cores, Fudge et al. (2016) highlighted the variability of the accumulation-temperature scaling relationship at the WDC site, noting three distinct periods in the 31-kyr record: an initial positive relationship from 31-15 ka (5.7% °C$^{-1}$), a weak negative relationship from 15-8 ka (-2.2% °C$^{-1}$), and a strong positive relationship from 8-0 ka (17.0% °C$^{-1}$). This suggests that the WDC site experiences synoptic-scale variability in precipitation not present at the EAIS ice core locations, and also not reproduced by the TraCE-21ka simulation. The $DG_{ns}$ simulation shows higher variability in accumulation as West Antarctic temperatures approach near-PI levels in the mid-Holocene (Fig 5c). We have no accumulation record for the JRI site to compare with the models. However, both simulations exhibit similar behavior in the AP region, with a shift in the scaling relationship with warming temperatures (Fig 5d). TraCE-21ka shifts to a slight negative relationship, while $DG_{ns}$ begins to show a weaker relationship with high variability, suggesting that the AP region is also impacted by dynamical changes in the model simulations that counters the thermodynamic temperature scaling. In the case of TraCE-21ka, this circulation change reduces the amount of precipitation over this region.

We also examine the ratio of the 500 yr precipitation change to the 500 yr temperature change to
explore how this relationship temporally evolves in the climate models (Fig 5e-h). Since some of the Antarctic
ice core accumulation reconstructions are not fully independent from the temperature reconstructions (Veres et
al., 2013; Frieler et al., 2015), analysis of the relationship cannot be performed for the ice cores except at multi-
millennial timescales (Fudge et al., 2016), hence only the climate models are considered here. Figure 5e-h
shows that the models remain nearly constant at the average scaling relationship, however, high variability
occurs in the mid-Holocene in TraCE-21ka in the coastal regions (yellow bars), and to a lesser extent in $DG_{ns}$,
suggesting a greater influence of synoptic-scale processes in precipitation at this time. The onset of high
variability in accumulation-temperature scaling occurs earlier in $DG_{ns}$ over the AP than over the coastal EAIS,
but it occurs synchronously over the two regions at ~9.5 ka in TraCE-21ka (Fig 5h,f). The WAIS and EAIS
regions exhibit comparatively less variability in the scaling relationship (Fig 5e,g), but the relationship is not
constant in time for either region.

**3.4 Ocean temperatures**

Both models simulate higher SSTs throughout the Southern Ocean in 6.5 ka relative to 18 ka, with the
largest increase occurring ~50°S (Fig 6a,b). SSTs generally increase by more over this the deglacial period in
$DG_{ns}$ than in TraCE-21ka, particularly in the Indian Ocean sector, southwest of New Zealand, and to the east of
southern South America, where SST increases of >5°C between 18 and 6.5 ka are observed in $DG_{ns}$. The models
show reasonable agreement with the SST change for the same period estimated in the marine proxy record, with
differences of <1.0°C at four of the six sites (Fig 6a-h). The mismatches occur at site MD03-2611 (south
Australia) and site TN057-13 (East Atlantic), where the models underestimate and overestimate the SST change,
respectively.

In terms of the absolute SSTs, the austral summer SSTs of the model generally show better agreement
with the SSTs reconstructed from marine sediments (high end of color shading in Fig 6c-h). $DG_{ns}$ exhibits
higher austral SSTs than TraCE-21ka at five out of the six sites, and thus more closely matches the
reconstructed SSTs. The main exception occurs in the Holocene at site TN057-13, in which the austral summer
SST of the proxy record declines and converges with the TraCE-21ka austral summer SSTs. Temporally, the
SST records show the largest SST increases in the early deglacial (Fig 6c,d,f,h) and in the early Holocene (Fig
6c-h), however, the models show relatively lower increases over these time periods.

At latitudes higher than 62°S, SSTs in both the TraCE-21ka and $DG_{ns}$ simulations exhibit modest
warming, with similar increases in the Ross, Amundsen and Bellingshausen Seas (Fig. 7b-e). The $DG_{ns}$
simulation shows greater sea surface warming in the Weddell Sea and the East Antarctic coastal seas (0.3-
0.6°C). In the subsurface, TraCE-21ka initially shows much cooler temperatures during the LGM relative to PI.
As a result, temperatures at 400 m depth increase by 2.1-2.7°C between 18 and 6.5 ka. In comparison, the $DG_{ns}$
simulation, which is much closer to its PI temperature at 18 ka, shows relatively modest warming during the
simulation period.

Meltwater input in the Southern Ocean at the time of MWP1A decreases AABW formation and
enhances the incursion of Circumpolar Deep Water on the Antarctic shelf in climate model simulations
(Menviel et al., 2010), thus leading to a sub-surface warming along the Antarctic coasts. Following the different
meltwater forcings used in the two models (see Methods), this sub-surface warming occurs earlier and for a

shorter duration in the TraCE-21ka than in DG$_{ns}$, with an offset of 1.2 kyr in the Weddell Sea (Fig 7c). SSTs show the opposite response in both models to this meltwater forcing. With the exception of MWP1A, the greatest amount of sub-surface warming in TraCE-21ka occurs between 11.5 and 9.5 ka (Fig 7a-e). DG$_{ns}$ shows also shows pronounced warming during this time interval along the East Antarctic coast (Fig 7d,e), but relatively modest temperature changes along the West Antarctic coast.

Considering the zonal perspective (64°S latitudinal band), the models display some similarities in the deglacial evolution of ocean temperatures to the depths of modern-day grounding lines (Fig 8). In response to the forcing of MWP1A, warmer ocean temperatures relative to PI are simulated in both models in the eastern Ross Sea to the Bellingshausen Sea, though at different depths (Fig 8c,i). In TraCE-21ka the warming is more pronounced and shallower (~100-200 m), with temperatures remaining higher than PI into the Holocene (Fig 8a-f).

f). In DG$_{ns}$, the warmer-than-PI temperatures occur at deeper depths, in the layer between 250 and 800 m (Fig. 8g-l). By 8 ka, much of the surface ocean exhibits temperatures similar to PI temperatures (Fig 8l). In contrast, TraCE-21ka shows slightly cooler temperatures, with the exception of the Amundsen and Bellingshausen sector at ~150 m depth, where temperatures remain higher than PI (Fig 8f).

**3.5 Sea ice**

       Both deglacial simulations show a substantial decrease in sea ice extent, thickness, and coverage in each sector between 18 and 6.5 ka, interrupted only by an increase in sea ice during the ACR (Fig 9,10). In comparison to the austral winter LGM sea ice extent and concentration reconstructions (Gersonde et al., 2005), DG$_{ns}$ austral winter sea ice extent (15% coverage) at 18 ka is consistent with the proxy-estimated LGM austral

winter sea ice extent in the Indian and Pacific Ocean sectors, but DG$_{ns}$ may overestimate sea ice extent in the eastern Atlantic sector (Fig 9a). TraCE-21ka is also consistent with the proxy record in the Indian and Pacific sectors, but overestimates the austral winter sea ice extent in the western Atlantic sector (Fig 9b). It should be noted that present-day simulations using the same GCM yield more extensive sea ice cover than observed (Yeager et al., 2006).

The models are consistent in terms of the behavior and absolute values of sea ice thickness and coverage (%) in each sector (Fig 10). The exceptions are the Weddell Sea and along the EAIS coast, in which TraCE-21ka produces sea ice that is approximately double in thickness. In contrast, thicker ice is observed in DG$_{ns}$ along the WAIS coast in the Ross, Amundsen and Bellingshausen Sea sectors. DG$_{ns}$ also exhibits higher centennial-scale variability in sea ice thickness, particularly in the Ross Sea sector. Mean 100 yr-average sea ice

coverage (%) decreases in the TraCE-21ka simulation by 22, 23, 25 and 9% in the Ross Sea, the Amundsen and Bellingshausen Seas, along the EAIS coast, and the Weddell Sea, respectively, through this interval (Fig 109e-h). The DG$_{ns}$ simulation shows similar sea ice coverage decreases of 24, 21, 30, and 12% in these coastal regions, respectively.

       Similar to the behavior observed with ocean temperatures, TraCE-21ka displays a shorter, but higher

magnitude response to Southern Ocean meltwater forcing associated with MWP-1A. The TraCE-21ka austral winter sea ice extent at 14 ka reaches lower latitudes than at 18 ka in the Atlantic sector (Fig 9b), and each coastal zone exhibits a short-lived increase in coverage and thickness (Fig 10). In contrast, DG$_{ns}$ maintains relatively higher sea ice thickness and coverage following the meltwater forcing for a longer duration (~1-1.5 ka), but with higher variability (Fig 10). Another difference is the increase in sea ice coverage in the EAIS

coastal sector in $DG_{ns}$ in the early Holocene (~12-11 ka), which is not observed in TraCE-21ka. $DG_{ns}$ also shows increases in sea ice coverage in the Ross, Amundsen and Bellingshausen sectors at this time, though with higher variability.

**3.6 Southern Ocean-Antarctic Climate connections**


As described in section 3.1, the regions displaying the greatest increases in continental surface temperature that are not associated with changing ice sheet topography occur along the continental margins (Fig 2a,b). While many factors, including the changes in greenhouse gas content, orbital forcing and meridional heat transport, contribute to these surface air temperature changes, albedo-driven radiative changes associated with
changes in sea ice coverage may also influence this enhanced warming observed in coastal areas. Surface albedo between 60-70°S over the Southern Ocean decreases by 0.18 and 0.15 in $DG_{ns}$ and TraCE-21ka, respectively. South of 60°S, TraCE-21ka shows larger decreases in surface albedo over the Ross, Amundsen and Bellingshausen Seas, whereas $DG_{ns}$ shows larger decreases over the Weddell Sea and along the eastern Antarctic coast. However, these sector differences are relatively small between the two models at these latitudes (<0.05).
Strong negative correlations exist in both models between surface air temperature, surface albedo and sea ice coverage in both models over the Southern Ocean (Fig 11). In both cases, the correlations between surface air temperature and sea ice coverage are weakest in the Weddell Sea, where the decrease in sea ice coverage is relatively lower. TraCE-21ka also shows a small area of positive correlation adjacent to the Antarctic Peninsula (i.e., warmer surface temperatures associated with higher sea ice coverage, and vice versa).
In addition to sea ice coverage, the surface albedo parameterization of both models accounts for the state of the sea ice surface (i.e., freezing or melting, snow coverage; Briegleb et al., 2004; Goosse et al., 2010). The TraCE-21ka simulation also accounts for the snow depth and snow age. Snow depth and surface melt also exhibit strong correlations to the surface air temperature over the Weddell Sea in TraCE-21ka, which may account for the difference in correlation with surface air temperature between surface albedo and sea ice coverage
(Supplementary Information).

In addition to linkages with continental temperatures, Southern Ocean conditions exert a strong influence on Antarctic accumulation patterns (Delaygue, 2000; Stenni et al., 2010). Figures 12 and 13 show the Pearson linear cross-correlation coefficients (r) of modelled decadal SST and precipitation for 4 ice core locations in each continental region (i.e., WDC, EDC, JRI, and Law Dome (LD)) for $DG_{ns}$ and TraCE-21ka,
respectively. SSTs generally show strong spatial correlations to continental accumulation in both models in the early deglacial period (18-12 ka), however, these relationships weaken or become more negative in the early to mid-Holocene (12-6.5 ka). The difference in millennial-scale variability between the two periods, which is higher in the early deglacial period, with Heinrich Stadial 1, the ACR, and Younger Dryas, likely contributes to this shift in correlation strength.
The SST-continental precipitation correlations are higher for TraCE-21ka than $DG_{ns}$ at every site except JRI, where the difference between the early deglacial period and early to mid-Holocene is starkest as the correlations transition from positive to slightly negative (Fig 13g,h). The models both exhibit high positive correlations between SSTs in the Indian and Pacific sectors with EDC precipitation in the early deglacial period (18-12 ka). Both models also show high positive correlations between WDC precipitation and SSTs in the
Pacific sector in the early deglacial period, but these correlations weaken in the Holocene, particularly in the

DG$_{ns}$ simulation (Fig 12c). Less model consistency exists at the coastal sites, most notably at LD (Fig 12c,d, 10c,d), which is likely related to the above-mentioned resolution limitation of DG$_{ns}$ along the EAIS coast.

To better understand the transitions in the accumulation relationships to continental temperatures and SSTs in the early Holocene, we consider changes in 500hPa geopotential height anomalies over the Southern Ocean and their relation to changes in atmospheric circulation. More specifically, the Amundsen Sea Low, which is the largest influence on meridional circulation in the region, acts as a major control on the temperature and precipitation in West Antarctica (Raphael et al., 2016). Modeling studies and ice core analyses have suggested that atmospheric teleconnections driven by tropical SST anomalies in the Pacific trigger a quasi-stationary wave response that reduces pressure over the Amundsen Sea. However, they offer conflicting accounts of teleconnection strength during glacial conditions (Timmermann et al., 2010; Jones et al., 2018). At 18 ka, the models display negative 500hPa geopotential height anomalies over much of the Southern Ocean (Fig 14), in contrast to the LGM simulation of Jones et al., (2018), which shows positive anomalies over the Amundsen Sea associated with a more El Niño-like state due to the orographic effects of the North American ice sheet.

The 500hPa geopotential height anomalies are more extreme in the TraCE-21ka than in DG$_{ns}$, particularly with regard to the positive anomalies observed over the ice sheet, which is primarily driven by the ice sheet topographic changes. Strong negative anomalies over the Amundsen Sea are observed in both models during the ACR (Fig 14c,j). A deepened Amundsen Sea Low is generally associated with enhanced northerly flow across the AP, allowing for intrusions of marine air masses over WAIS (Hosking et al., 2013). Through time, the simulated geopotential height increases over the Southern Ocean to near-PI levels, reaching a state of reduced variance over the Amundsen Sea in the early Holocene, consistent with the timing of reduced accumulation-temperature scaling over the AP in both model simulations (Fig 5d).

## 4. Discussion

### 4.1 Regional patterns of deglacial climate evolution

The deglacial model simulations and Antarctic ice core records demonstrate clear regional differences between East and West Antarctica as well as between coastal regions and the continental interior in terms of deglacial warming, driven primarily by oceanic influences. In comparison to the East Antarctic ice cores, the WDC isotope record indicates that the glacial termination of the more heavily marine-influenced WAIS initiated ~2 kyr earlier than EAIS, with warming that reached near-mid Holocene levels by 15 ka, coincident with a decline in circum-Antarctic sea ice (WAIS Divide Project Members, 2013; Cuffey et al., 2016). Although the climate models underestimate the rate of warming between 18-15 ka over WDC, they do show greater warming rates for the WAIS, AP, and coastal EAIS regions, also occurring synchronously with the greatest declines in coastal sea ice, as compared to the less-marine influenced EAIS interior. The ice cores and models constrain centennial-scale warming rates to within 0.6°C, with the most rapid warming occurring at the end of the ACR, excluding the abrupt ice mask-induced warming in TraCE-21ka.

In terms of the spatial pattern of temperature change from the termination to the mid-Holocene, the greatest continental warming occurs over sea ice-adjacent land and areas of lowered ice surface elevation, highlighting the strong effects of albedo and orography on regional temperatures in Antarctica. The sea ice-albedo effect may be overestimated in the models over the AP, as demonstrated by the JRI temperature

reconstruction; however, Mulvaney et al. (2012) do note the strong sensitivity of AP temperature to ice shelf-related changes, with rapid warming events in the early Holocene and over the past 100 years ($1.56 \pm 0.42°C$). Additional temperature and methane sulphonic acid reconstructions of coastal Antarctic ice cores could further elucidate the regional sensitivities to localized ice shelf and sea ice changes and identify possible thresholds.

In comparison to surface temperature, the regional evolution of accumulation and its relationship to surface temperature and conditions in the Southern Ocean through the deglacial period remain less clear. The two climate simulations show substantial differences with regard to the overall magnitude of accumulation changes in each region. In both the ice core records and the models, accumulation over the ice sheet generally shows higher variability in coastal regions, as expected, with the EAIS interior displaying consistent temperature-accumulation scaling relationships at various sites, as previously demonstrated by Frieler et al. (2015). A primary limitation of the intermediate complexity LOVECLIM model is the inadequate resolution for the steep elevation gradient of the coastal EAIS (Goosse et al., 2012), which leads to a precipitation bias in the region in the $DG_{ns}$ simulation. This effect worsens through time due to the reduced moisture gradient between the Southern Ocean and the EAIS interior, as P-E increases over the EAIS interior, presumably from thermodynamic scaling with temperature.

The climate models show limited variability in accumulation-temperature scaling over WAIS, in contrast to the WDC record (Fudge et al., 2016); however, the AP exhibits a marked shift in the early to mid-Holocene, which may be due to changes in atmospheric circulation. This climatic shift in the early to mid-Holocene coincides with the reduced variance and weakening of the Amundsen Sea Low, which achieves near-present day conditions in both models by ~12 ka. The weakened Amundsen Sea Low appears to reduce the accumulation-temperature scaling over the AP in both models, and in the case of TraCE-21ka, a slight negative scaling relationship is observed. Although this is not observed over the greater WAIS region, it should be noted that the timing is roughly consistent with the slight negative scaling period observed in the WDC record (15-8 ka).

It may be expected that the retreat of sea ice and increased area of open ocean may introduce additional moisture sources, thereby enhancing precipitation relative to temperature. In fact, coastal ice core records of the Holocene, including Taylor Dome, Law Dome, Siple Dome, have exhibited decoupling between accumulation and temperature that has been attributed to moist-air cyclonic activity and changes in sea ice conditions (Monnin et al., 2004). Although the climate simulations do not exhibit a substantial increase in the scaling relationship with reduced sea ice coverage, much higher temporal variability is observed over coastal EAIS in the mid-Holocene. Likewise, the reduced correlations between Antarctic precipitation at the inland ice core locations and SSTs in the Holocene may demonstrate that this effect is not limited to the coasts in these climate simulations.

The linkages between sea ice coverage, atmospheric dynamics and continental accumulation may also have important implications for future projections of the more heavily marine-influenced WAIS climate, as GCMs used in the Coupled Model Intercomparison Project 5 (CMIP5) tend to project higher accumulation-temperature scaling in the future due to reduced sea ice coverage (Palerme et al., 2017). This effect may also be quite regionally dependent, as no statistically significant increase has been observed over the Thwaites glacier over the past three decades (Medley et al., 2013). The results here also highlight that dynamical processes cannot be discounted in Antarctic accumulation projections. Additional moisture budget analysis is warranted to better constrain the effects of circulation changes and cyclonic-driven precipitation on Antarctic accumulation in

transient climate simulations, particularly in relation to tropical and mid-latitude teleconnections with the Amundsen Sea Low.

**4.2 Regional patterns of deglacial ocean evolution**

Given the lack of Southern Ocean proxy records and the large uncertainties of those that do exist, the current understanding of the deglacial evolution of Southern Ocean conditions remains limited. The available SST records from this region that cover the deglacial period generally show two key phases of rapid warming,
that is, at the start of the deglaciation (~18-14.5 ka) and following the ACR into the early Holocene (~12.5-10 ka; Calvo et al., 2007). However, differences between the records can be observed, even between marine records located in close proximity, but at different latitudes; for example, cores ODP-1233 and MD07-3128 off the coast of southern Chile display differences in the timing of SST warming (Caniupan et al., 2011). Also different among the records is the presence and duration of Holocene cooling, with an early and long-term cooling trend
observed in TN057-13 (Anderson et al. 2009), but no such trend observed in TN057-6 (Anderson et al. 2014). Both records are from the East Atlantic, but on opposite sides of the modern Antarctic Polar Front. The SSTs are also based on different proxies, hence the divergent behaviour may be driven by out-of-phase trends of seasonal insolation during the Holocene (Leduc et al., 2010).

Using SST reconstructions to constrain the model results is challenging considering the spatial and
temporal gaps in the records, uncertainties in the age scales, and the potential for proxy-related biases. The proxies may also be recording temperatures at depths other than the surface in accordance with the depth at which the diatoms and foraminfera occurred. The modelled austral summer SSTs generally show better agreement with reconstructed SSTs rather than the modelled annual mean SSTs averaged over 100 yr. We focus on decadal- to centennial-scale changes because the TraCE-21ka ocean output are available as decadal averages,
however, we do note that the modelled annual variability of SST in $DG_{ns}$ is higher, with average austral summer SST ranges of 4.25°C in the early deglacial (18-17 ka) and 3.27°C in the middle Holocene (7.5-6.5 ka) at the 6 proxy record sites. It is therefore expected that the modelled SSTs do have greater overlap with the proxy reconstructions on an annual-scale, and are likely within the large range of proxy uncertainty (e.g., Tierney and Tingley, 2018). In terms of the temporal evolution, the models do not reproduce a decreasing trend in the mid-
Holocene at any of the analysed sites. While this may in part be driven by biases in particular records, since not every SST reconstruction displays this trend, it has also been posited that this Holocene model-data discrepancy is the result of model bias in the sensitivity to decreasing obliquity, with the positive albedo feedback with snow, ice, and vegetation too weakly represented (Liu et al., 2014). Lastly, given that the locations of the marine proxy records are in close proximity to modern ocean frontal systems, it should be noted that shifts in these
systems through the deglacial period may account for some differences between the reconstructed and simulated ocean temperatures.

In addition to these challenges, there are a few known model biases with regard to the simulation of modern ocean circulation in both models that should be considered. Although the ocean component of LOVECLIM, which is used for the $DG_{ns}$ simulation, generally shows strong agreement with observations of
Southern Ocean overturning circulation, it produces shallower convection in the North Atlantic (Dreisschaert et al. 2005), which may have implications for ocean circulation over these millennial timescales. The CCSM3

model, which is used for TraCE-21ka, overestimates modern sea ice extent and thickness in both hemispheres, likely resulting from stronger-than-realistic wind forcing (Yeager et al., 2006). This also influences both ocean temperatures and salinities in the Ross and Weddell Seas, leading to AABW that is denser than observed (Danabasoglu et al. 2012). From a paleoclimate perspective, CCSM3 is an outlier in terms of sea ice and southern westerlies strength among climate models in the Paleoclimate Modelling Intercomparison Project 2 (PMIP2) for the LGM (Rojas et al., 2009). It should be noted that more recent versions of these models have improved on these issues, with CCSM4 showing better agreement to the LGM sea ice proxy data of Gersonde et al. (2005) in the Southern Ocean (Brady et al., 2013). Therefore, it is expected that if this transient experiment was performed using an updated version, better model-proxy agreement with regard to sea ice would be observed.

Mesoscale eddies are ubiquitous in the Southern Ocean and the eddy-driven circulation counteracts the Eulerian component driven by Ekman transport. Coarse resolution climate models are not able to resolve mesoscale eddies and instead use the Gent and McWilliams parametrization (Gent and McWilliams, 1990; Gent et al., 1995). While previous studies suggested that, due to their coarse resolution, the climate models used here could have a stronger-than-realistic sensitivity to wind forcing (Farneti and Delworth, 2010), more recent studies suggested that the Antarctic Circumpolar Current and Southern Ocean overturning response to wind and freshwater forcing in coarse and high-resolution models was in general agreement (Downes and Hogg, 2013; Menviel et al., 2015; Menviel et al., 2018). Nevertheless, a more detailed analysis of the relationship between wind changes and Southern Ocean circulation is required to further explore this possibility. In particular, reconstructions of westerly wind strength and position (e.g., Kohfield et al., 2013) paired with reconstructions of Southern Ocean circulation (e.g., Rae et al., 2018) could offer useful constraints for future paleoclimate simulations.

One of the larger unknowns of the last deglaciation that is of great consequence to Antarctic ice sheet retreat history is the temporal evolution of coastal ocean temperature changes at grounding line depths. This is particularly important for WAIS, as the reverse slope bathymetry of the continental shelf creates a configuration prone to retreat, a process that may be currently impacting some regions (Pritchard et al 2012; Favier, 2014). Although the seafloor geometry likely acted as a first order control on ice sheet retreat of grounded ice in some locations (The RAISED Consortium, 2014; Halberstadt et al., 2016), changes in ocean temperature and warm water intrusions underneath ice shelves can dictate the timing of the retreat. The proxy record provides few constraints on coastal sub-surface ocean temperature changes, but the deglacial climate model simulations analysed here do offer some insights. Both models show differences between ocean temperature evolution in the sub-surface as compared to the surface, suggesting that SST reconstructions may not be useful for understanding past ocean forcing of the Antarctic ice sheet. This difference is more pronounced in TraCE-21ka, which shows substantially cooler glacial ocean temperature anomalies in the sub-surface as compared to the surface close to the continent.

Despite differences in the prescribed meltwater forcings associated with MWP1A, both TraCE-21ka and DG$_{ns}$ show pronounced sub-surface warming following this event. The input of meltwater in the Ross and Weddell Seas at the time of MWP1A decreases AABW formation and enhances the incursion of Circumpolar Deep Water on the Antarctic shelf (Menviel et al., 2011). The warming is most evident in the Amundsen and Bellingshausen Sea sector in both models. Ice sheet model simulations using various climate forcings, including

$DG_{ns}$ output, have suggested that the initial retreat of WAIS occurred in the AP region and outer Weddell Sea, which is consistent with this location of sub-surface ocean warming (Golledge et al., 2014). The meltwater forcing resulting from this retreat may have resulted in a positive feedback between sea ice, AABW production, and thermal erosion at the ice sheet grounding line.

**4.3 Transient climate modeling limitations**

The main limitations of these experiments involve the poor paleo-constraints with regard to certain boundary conditions. Both transient simulations display discrepancies with the paleo-records across the ACR and early Holocene. In the case of TraCE-21ka, the ACR is more abrupt and of a larger magnitude than in the ice core temperature and SST reconstructions, which is related to how the MWP1A meltwater forcing is applied between 14.35 and 13.85 ka (see Methods).

He (2011) performed a sensitivity experiment with CCSM3, in which the prescribed meltwater discharge in both hemispheres associated with MWP1A was excluded, that improves on the ACR model-proxy agreement. The meltwater discharge applied to the North Atlantic prior to the ACR (17-14.67 ka) suppresses the upper cell of the AMOC. When this freshwater forcing is terminated at 14.67 ka, AMOC recovers. Without the inclusion of MWP1A meltwater discharge during the Bølling-Allerød interstadial, Antarctic surface temperatures can remain cooler for a longer duration due to the bipolar seesaw response to the rapid increase in AMOC strength, which leads to a warming in the Northern Hemisphere. Pedro et al. (2015) note that the cooling in this ACR simulation induced by the AMOC strengthening at the end of Heinrich Stadial 1 is further amplified by the expansion of Southern Hemisphere sea ice. However, the ACR simulation is also limited in that it does not capture Older Dryas cooling over the Northern Hemisphere (He, 2011), and the improved ACR representation without freshwater forcing is difficult to reconcile with sea level records that indicate the pronounced meltwater discharge of MWP1A (Deschamps et al., 2012).

In comparison to TraCE-21ka, the $DG_{ns}$ experiment better matches the duration of the ACR, but shows more modest temperature changes, particularly over the EAIS interior region. Prescribed Southern Hemisphere meltwater helps induce the ACR in $DG_{ns}$ by reducing AABW formation, thereby decreasing the southward oceanic heat transport (Menviel et al., 2011). The longer duration and lower magnitude of meltwater forcing applied to the Ross and Weddell Seas to represent MWP1A in $DG_{ns}$ likely drives these differences with TraCE-21ka, which shows a shorter and more pronounced ACR. The shutdown of the AMOC at ~13 ka associated with the Younger Dryas and AABW strengthening lead to a significant warming at high southern latitudes in the $DG_{ns}$ experiment. However, this warming does not continue into the early Holocene (11.5-10 ka) as indicated by the Southern Ocean SST proxy records. TraCE-21ka similarly does not reproduce the warming through this interval.

Bereiter et al., (2018) suggest that this model-proxy mismatch during the early Holocene is not uncommon among GCMs and intermediate complexity models, and may indicate that the current generation of climate models may underestimate ocean heat uptake or that the understanding of the boundary conditions used for ACR/Younger Dryas climate simulations may be flawed. More specifically, given that the precise timing, magnitudes and locations of MWP1A and MWP1B remain a subject of debate (Bard et al., 2010; Deschamps et al., 2012; Gregoire et al., 2016), yet the prescribed meltwater forcings have significant effects on the climate

evolutions in these transient simulations (He, 2011; Menviel et al., 2011), accurate model representation through this interval may remain limited until these meltwater discharge events are better constrained.

Additional transient climate simulations of the deglaciation using different GCMs and intermediate complexity models are required to more clearly determine which of these differences and similarities are model dependent. Considering the significance but high uncertainty of the prescribed meltwater forcings in these simulations, the incorporation of coupling between GCMs and dynamic ice sheet models is a necessary step to ensure that climatically relevant ice-ocean interactions and feedbacks, such as sub-ice shelf melting and ice berg calving, are adequately reproduced in future transient climate simulations. Marine sediment and coastal ice core reconstructions that can be used to evaluate these climate model performances in terms of coastal ocean circulation and sea ice would also be particularly helpful in better understanding Antarctic climate evolution and ice sheet retreat history during the deglaciation.

## 5. Summary and Conclusions

Based on spatial and temporal analysis of two transient climate simulations of the last deglaciation, one using a fully coupled GCM (TraCE-21ka) and one using an intermediate complexity model (LOVECLIM $DG_{ns}$), we explore the regional aspects of Antarctica's deglacial climate evolution. We also assess climate model performances with regard to model output most relevant to ice sheet model simulations, including surface temperature, surface mass balance, coastal ocean temperatures, and sea ice. The main findings of this study are as follows:

The greatest continental surface temperature warming from the glacial termination (18 ka) to the mid-Holocene (6.5 ka) occurs along coastal margins and, in the case of TraCE-21ka, regions with the greatest decrease in ice surface elevation, suggesting the importance of sea ice-albedo feedbacks and ice sheet dynamics in Antarctica's deglacial evolution. The centennial-scale rates of temperature change are reasonable as compared to the proxy record.

Strong discrepancies in modelled accumulation (P-E) are observed in the two climate simulations, particularly along the EAIS coast, which are related to the coarser resolution of the atmosphere component of $DG_{ns}$ compared to TraCE-21ka. TraCE-21ka successfully captures the magnitude of accumulation change estimated for WDC and is within proxy uncertainty of EDC, whereas $DG_{ns}$ underestimates the accumulation change at WDC and overestimates the change at EDC. Accumulation-temperature scaling relationships are also not spatially or temporally uniform, with the coastal EAIS and AP showing higher variability in the relationship in the early to mid-Holocene.

The models show some substantial differences with regard to coastal SSTs and sub-surface ocean temperatures. The TraCE-21ka simulation exhibits considerably cooler glacial Southern Ocean temperature anomalies, and as a result, experiences a greater degree of warming to the mid-Holocene. Both models show sub-surface warming in response to Southern Ocean meltwater forcing of MWP1A that is most pronounced between the Amundsen and Bellingshausen Sea sector, though this warming occurs at different depths.

Both models reasonably capture glacial sea ice extent estimated in the proxy record, but may overestimate austral winter sea ice extent in parts of the Atlantic sector. The models are relatively consistent in terms of changes in sea ice coverage and thickness along the Antarctic coast. The exception is the Weddell Sea, in which TraCE-21ka shows considerably thicker sea ice.

Correlations between Southern Ocean SSTs and Antarctic accumulation weaken in the Holocene, coinciding with the reduction of sea ice coverage and increased variability in accumulation-temperature scaling over the coastal regions. A weakened and more stable Amundsen Sea Low appears to decrease accumulation-temperature scaling over the AP. A more detailed moisture budget analysis with an emphasis on tropical teleconnections to the Amundsen Sea Low is required to better understand this climatic shift.

The greatest model-proxy mismatch occurs with the temperature and accumulation changes associated with the ACR and the SST changes associated with the early Holocene ocean warming event, in which the models do not adequately capture the precise magnitude and duration of change. These mismatches may result from model bias in large-scale ocean circulation, biases in the proxy records, or poorly constrained boundary conditions during these time periods. In particular, deglacial climate evolution is highly sensitive to the timing, magnitude and location of prescribed meltwater forcing during this time interval.

Given the relatively limited number of transient climate simulations of the deglaciation, it remains challenging to assess model skill and identify model-dependent biases in capturing the transient aspects of climate, including the factors that impact climate-forced ice sheet model simulations. As such, the community would be well served by additional transient climate simulations using different climate models, the incorporation of dynamic ice sheet models in climate simulations, and high-resolution proxy records that can aid in climate model assessment.

**Acknowledgements**

We gratefully acknowledge the teams behind the TraCE-21ka and LOVECLIM DG$_{ns}$ experiments for producing and sharing model output, publicly available via the NCAR Climate Data Gateway and the Asia-Pacific Data Research Center, respectively. We also thank the Antarctic ice core and marine sediment proxy communities for the use of their data and two anonymous reviewers for their constructive comments. Funding for this project was provided by the New Zealand Ministry of Business, Innovation, and Employment Grants through Victoria University of Wellington (15-VUW-131) and GNS Science (540GCT32). DPL acknowledges support from the Antarctica New Zealand Doctoral Scholarship program. NG acknowledges support from the Royal Society Te Aparangi under contract VUW1501. LM acknowledges support from the Australian Research Council grant DE150100107.

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

**Figures & Tables**

**Table 1.** Model details and specifications.

| Model Simulation | Atmosphere Component | Land Component | Ocean Component | Sea Ice component |
|---|---|---|---|---|
| TraCE-21ka | Community Atmospheric Model 3 (CAM3) (Collins et al., 2006)<br><br>3.75° horizontal resolution<br><br>26 hybrid coordinate vertical resolution | Community Land Model-Dynamic Global Vegetation Module (CLM-DGVM) (Levis et al. 2004)<br><br>3.75° horizontal resolution | Parallel Ocean Program (POP) (Collins et al., 2006)<br><br>vertical z-coordinate with 25 levels,<br><br>3.6° longitudinal resolution, and variable latitudinal resolution, with finer resolution near the equator (~0.9°) | Community Sea Ice Model (CSIM) (Collins et al., 2006)<br><br>Thermodynamic/ dynamic model that includes sub-grid ice thickness distribution<br><br>3.6° longitudinal and variable latitudinal resolution, with finer resolution near the equator (~0.9°) |
| LOVECLIM DG$_{ns}$ | ECBilt (Opsteegh et al., 1998)<br><br>5.6° horizontal resolution<br><br>quasi-geostrophic T21 spectral, 3 levels | VECODE (Brovkin et al., 1997)<br><br>dynamic vegetation module<br><br>5.6° horizontal resolution | CLIO (Campin and Goosse, 1999)<br><br>primitive equations, z-coordinate,<br>3° horizontal resolution with 20 vertical levels<br><br>LOCH (Mouchet and Francois, 1996)<br><br>3-dimensional ocean carbon cycle model | Thermodynamic/ dynamic sea ice model coupled to CLIO |





**Table 2.** Antarctic and Southern Ocean proxy record details.

| Record name | Type | Measurement | Method | Location | Reference |
|---|---|---|---|---|---|

| | | | | | |
|---|---|---|---|---|---|
| *Vostok (V)* | Ice core | Temperature anomaly | $\delta$D temperature calibration (9.0‰ °C$^{-1}$) | East Antarctica, 78.5°S, 107°E | Lorius et al., 1995; Petit et al., 1999 |
| *Dome Fuji (DF)* | Ice core | Temperature anomaly | $d$ temperature calibration (7.7‰ °C$^{-1}$) | East Antarctica, 77°S, 39°E | Uemura et al., 2012 |
| *James Ross Island (JRI)* | Ice core | Temperature anomaly | $\delta$D temperature calibration (6.7+/-1.3‰ °C$^{-1}$) | Antarctic Peninsula, 64°S, 58°W | Mulvaney et al., 2012 |
| *EPICA Dome C (EDC)* | Ice core | Temperature anomaly; Accumulation | $\delta$D temperature calibration (7.6‰ °C$^{-1}$); Ice flow model | East Antarctica, 75°S, 124°E | Jouzel et al., 2007; Parennin et al., 2007; Stenni et al., 2010 |
| *WAIS Divide (WDC)* | Ice core | Surface temperature; Accumulation | Water stable isotope record, borehole temperatures and nitrogen isotopes (7.9‰ °C$^{-1}$); Ice flow model | West Antarctica, 79.5°S, 112°W | WAIS Divide Project Members, 2013; Cuffey et al., 2016; Fudge et al., 2016 |
| *Core MD03-2611* | Marine sediment core | SST | Alkenone-derived | Murray Canyons area, 36°S, 136°E | Calvo et al., 2007 |
| *ODP site 1233 core* | Marine sediment core | SST | Alkenone-derived | South Chile, 41°S, 74°W | Kaiser et al., 2005 |
| *Core MD97-2120* | Marine sediment core | SST | Mg/Ca-derived | Chatham Rise, 45°S, 175°E | Pankhe et al., 2003 |
| *Core TN057-6* | Marine sediment core | SST | Alkenone-derived | East Atlantic, 43°S, 9°E | Anderson et al., 2014 |
| *Core MD07-3128* | Marine sediment core | SST | Alkenone-derived | Magellan Strait, 53°S, 75°W | Caniupan et al., 2011 |
| *Core TN057-13* | Marine sediment core | Seasonal SST (Feb) | Diatom-derived | East Atlantic Polar Front, 52°S, 5°E | Anderson et al., 2009 |




**Table 3.** Change from 18 to 6.5 ka at Antarctic ice core sites (surface temperature, °C; accumulation, cm/yr) and marine sediment core sites (SST, °C) estimated in the proxy records and simulated in DG$_{ns}$ and TraCE-21ka for austral summer (December to February) and austral winter (June to August). Proxy records were linearly

interpolated to 100 year averages. Bold font indicates a match between the seasonal range of change in the

models and the proxy estimation at Antarctic ice core sites. Red font indicates that the seasonal range of change

in the model does not overlap with the uncertainty range of the ice core record. We use uncertainties of +30% to

-10% for V and DF temperature (Jouzel et al. 2003), +/- 2°C for EDC temperature (Stenni et al., 2010), +/-1.8°C

for WDC temperature (Cuffey et al., 2016), and +/-1.0°C for JRI temperature (Mulvaney et al., 2012). For

accumulation, we assume uncertainties of +/-25% (Fudge et al., 2016).

| Record | Proxy estimation (6.5 – 18 ka) | DG$_{ns}$ seasonal range (6.5 – 18 ka) | TraCE-21ka seasonal range (6.5 – 18 ka) |
|---|---|---|---|
| V | 8.04°C (7.24-10.45°C) | 2.75-6.25°C, 6.60-11.21 cm/yr | 9.02-10.40°C, 0.99-1.42 cm/yr |
| EDC | 8.18°C (6.18-10.18°C), 1.42 cm/yr (1.14-1.70 cm/yr) | **3.25-9.87°C**, 3.25-17.60 cm/yr | 8.99-10.12°C, 1.68-4.16 cm/yr |
| DF | 8.70°C (7.83 -11.31°C) | 3.70-8.23°C, 0.85-7.72 cm/yr | **8.33-10.69°C**, 3.58-4.09 cm/yr |
| WDC | 10.20°C (8.4-12.0°C), 11.8 cm/yr (8.85-14.8 cm/yr) | **8.40-11.68°C**, -5.00-8.11 cm/yr | 10.34-15.28°C, **11.37-19.91 cm/yr** |
| JRI | 6.26 (5.26-7.26°C) | 6.99-11.84°C, 10.07-21.25 cm/yr | **3.74-8.75°C**, 4.04-5.25 cm/yr |
| MD03-2611 | 7.77°C | 1.51-1.90°C | 2.26-3.05°C |
| ODP-1233 | 3.44°C | 2.90-4.02°C | 2.32-3.06°C |
| MD97-2120 | 3.40°C | 3.09-4.58°C | 1.64-2.52°C |
| TN057-6 | 5.69°C | 2.89-3.95°C | 2.03-2.44°C |
| MD07-3128 | 5.66°C | 4.68-5.08°C | 2.28-2.83°C |
| TN057-13 | 0.14°C | 2.97-3.49°C | 1.29-2.17°C |

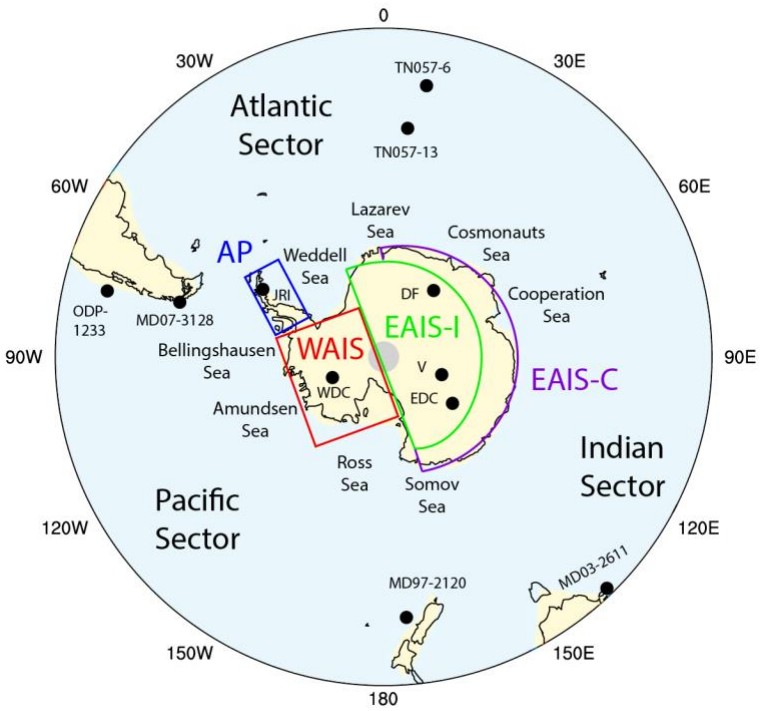


**Figure 1:** Polar-stereographic view of Antarctica and the Southern Ocean (maximum Latitude of 35°S). The colors indicate the land and ocean mask of the TraCE-21ka simulation (yellow and light blue, respectively). The continental regions, namely, the Antarctic Peninsula (AP), the West Antarctic Ice Sheet (WAIS), the East Antarctic Ice Sheet interior (EAIS-I), and the East Antarctic Ice Sheet coastal region (EAIS-C) are outlined in

the colored boxes (blue, red, green, and purple, respectively). The locations of the Antarctic ice core and marine sediment records used in this analysis are indicated by the black dots.

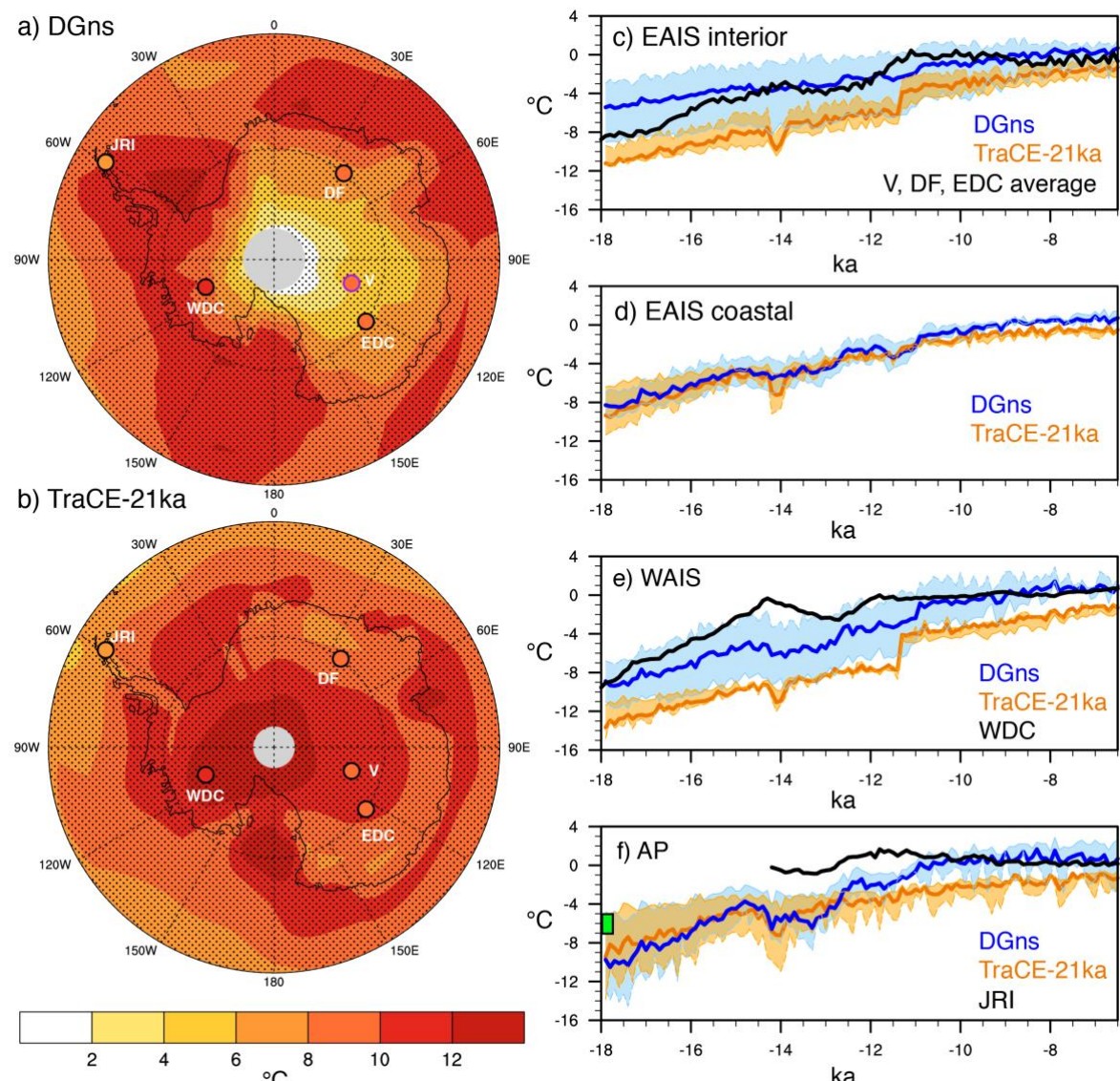

**Figure 2:** Surface temperature change (°C) as simulated in (a) DG_ns and (b) TraCE-21ka for the period of 18 to 6.5 ka (maximum Latitude of 60°S). Dotted lines indicate latitude (intervals of 15°) and longitude (intervals of 30°). Ice core locations of the JRI, WDC, DF, V, and EDC sites are marked by filled circles, with the fill color corresponding to the change estimated in the ice core record, black outlines indicating a match in warming between the ice core and model simulation (i.e., the ice core temperature change from 18 to 6.5 ka is within the range of seasonal temperature changes of the climate model), and green (purple) outlines indicating an overestimation (underestimation) in warming by the models. For these model-proxy comparisons, we use site-specific model averages of land surface grid cells: JRI (63-65°S, 59-62°W), WDC (77-82°S, 115-109°W), DF (75-79°S, 36-42°E), V (77-82°S, 104-110°E), EDC (73-77°S, 121-127°E). Stippling indicates a difference between decadal output of 18.0-17.5 ka and 7.0-6.5 ka that is significant at the 95% confidence level. (c-e) 100-yr average surface temperature time series (°C) of four regions (EAIS interior: 83°S-75°S, 30°W-165°E; coastal EAIS: 75°S-68°S, 15°W-165°E; WAIS: 83°S-72°S, 165°E-30°W; AP: 72°S-64°S, 64°W-59°W) relative to PI of both model simulations and ice core temperature reconstructions of sites located therein (DG_ns in blue, TraCE-21ka in orange, ice cores in black). The colored shading indicates the seasonal range calculated from the 100-yr average austral summer and winter temperature anomalies. For EAIS, the ice core reconstruction is an

average of the DF, V and EDC sites. The green box in (f) indicates the LGM temperature anomaly estimated for
JRI (6.01±1.0°C; Mulvaney et al., 2012).

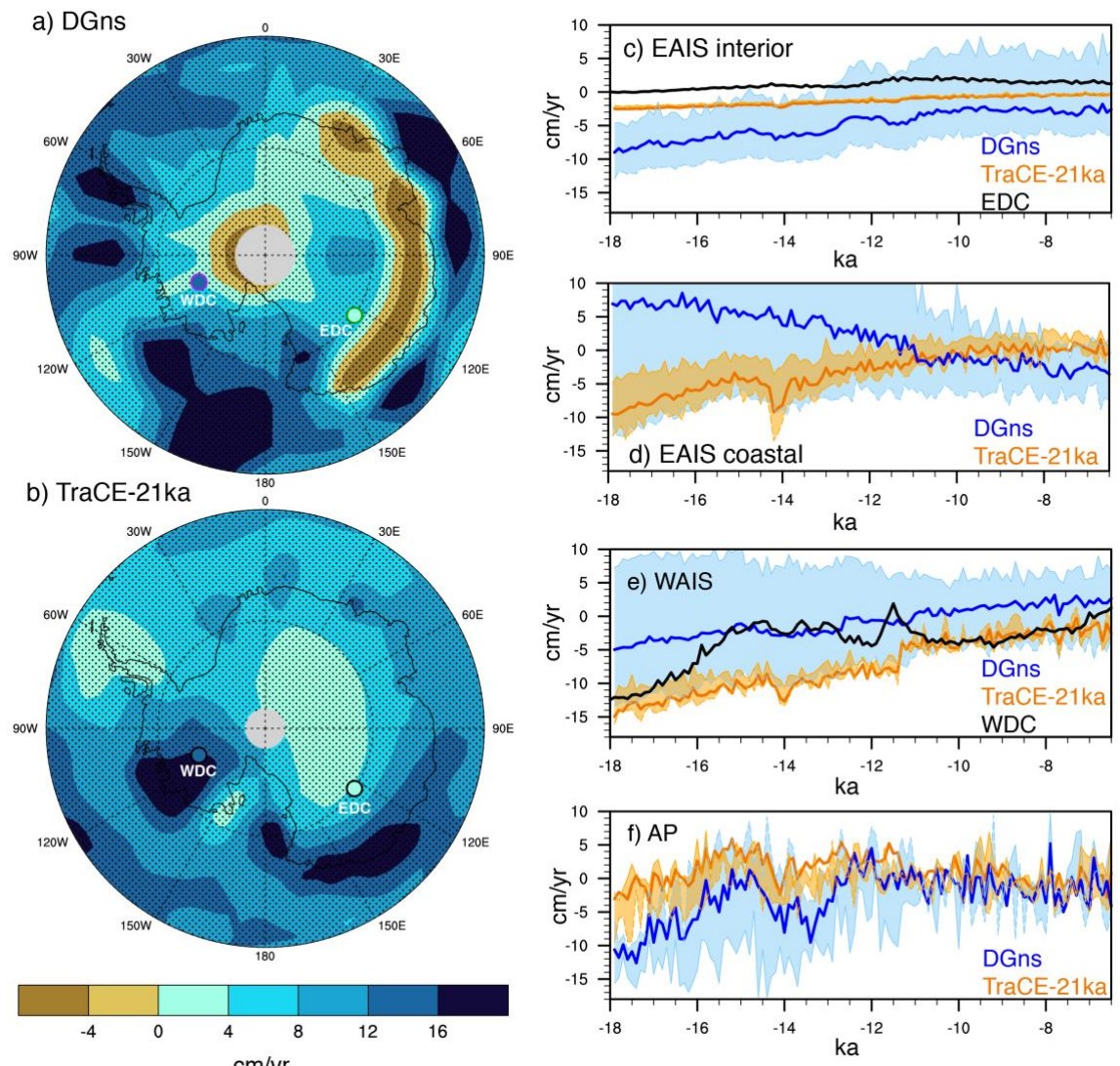

**Figure 3:** Precipitation change (cm/yr) as simulated in (a) DG$_{ns}$ and (b) TraCE-21ka for the period of 18 to 6.5
ka (maximum Latitude of 60°S). Dotted lines indicate latitude (intervals of 15°) and longitude (intervals of 30°).
Ice core locations of WDC and EDC are marked by filled circles, with the fill color corresponding to the change
estimated in the ice core record, black outlines indicating a match in warming between the ice core and model
simulation (i.e., the ice core temperature change from 18 to 6.5 ka is within the range of seasonal temperature
changes of the climate model), and green (purple) outlines indicating an overestimation (underestimation) in
warming by the models. For these model-proxy comparisons, we use site-specific model averages of land
surface grid cells: WDC (77-82°S, 115-109°W), EDC (73-77°S, 121-127°E). Stippling indicates a difference
between decadal output of 18.0-17.5 ka and 7.0-6.5 ka that is significant at the 95% confidence level. (c-f) 100-
yr average accumulation time series (cm/yr) of four regions (EAIS interior: 83°S-75°S, 30°W-165°E; coastal
EAIS: 75°S-68°S, 15°W-165°E; WAIS: 83°S-72°S, 165°E-30°W; AP: 72°S-64°S, 64°W-59°W) relative to PI

of both model simulations (P-E) and ice core accumulation reconstructions of sites located therein (DG$_{ns}$ in blue, TraCE-21ka in orange, ice cores in black). The colored shading indicates the seasonal range calculated from the 100-yr average austral summer and winter temperature anomalies.

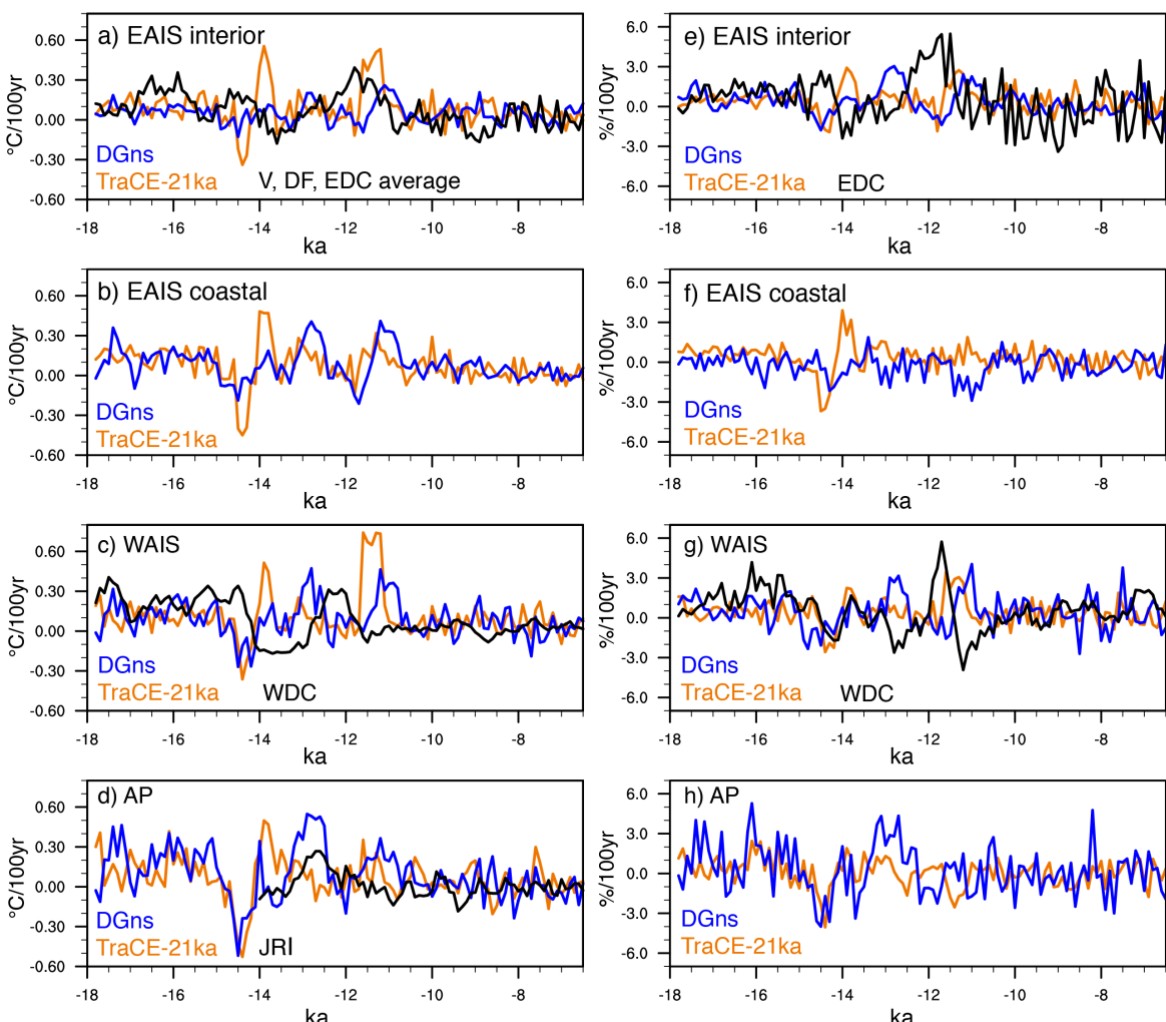

**Figure 4:** Centennial-scale rates of (a-d) surface temperature change (°C) and (e-h) relative accumulation change (%; DG$_{ns}$ in blue, TraCE-21ka in orange, ice cores in black).

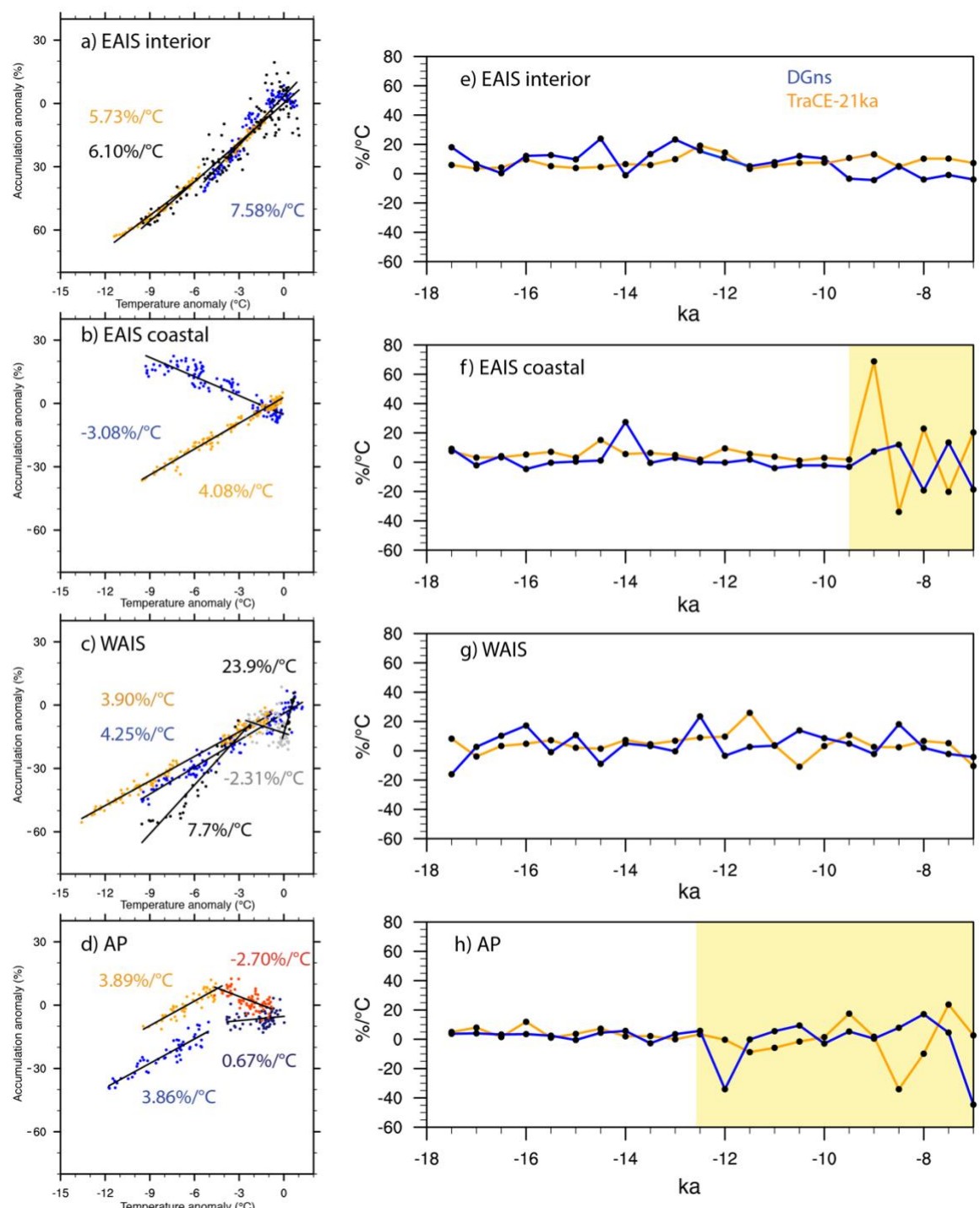

**Figure 5:** (a-d) Scaling relationships of accumulation (% relative to PI) and temperature (°C relative to PI) in each region. Black and grey dots refer to the proxy record, blue and purple dots refer to the DG$_{ns}$ simulation, and orange and red dots refer to the TraCE21ka simulation. (e-h) The ratio of the change in precipitation (%) to the change in temperature (°C) per 500 years. The yellow bars in panel f and h indicate a shift to higher variability in accumulation-temperature scaling in the AP and EAIS coastal regions.

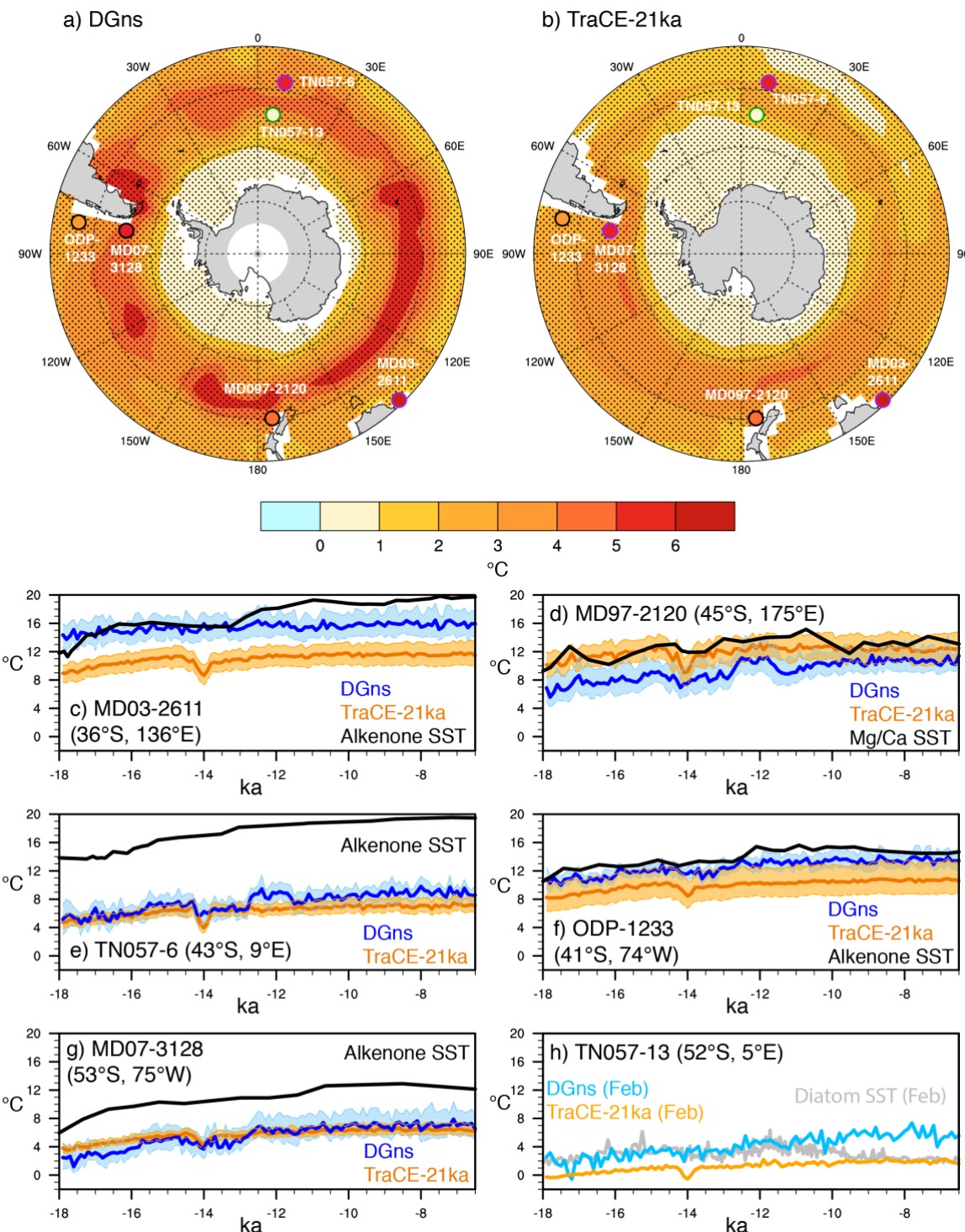

**Figure 6:** Sea surface temperature (SST) change (°C) as simulated in (a) DG$_{ns}$ and (b) TraCE-21ka for the period of 18 to 6.5 ka (maximum Latitude of 35°S). Dotted lines indicate latitude (intervals of 15°) and longitude (intervals of 30°). Marine sediment locations are marked by open circles, with black outlines indicating a match in warming between the ice core and model simulation (i.e., the estimated proxy SST change from 18 to 6.5 ka is within the range of seasonal SST changes of the climate model), and green (purple) outlines indicating an overestimation (underestimation) in warming by the models. Stippling indicates a difference between decadal output of 18.0-17.5 ka and 7.0-6.5 ka that is significant at the 95% confidence level. (c-h) Time

series of 100-yr average mean annual SST from the models and SST proxy reconstructions (°C) at each

individual marine sediment core site. The color shading represents the seasonal range calculated from the 100-yr average austral summer and winter temperature anomalies. In panel h, only a seasonal (February) proxy reconstruction is available; therefore, we only show modelled February SSTs from the climate models for this site.


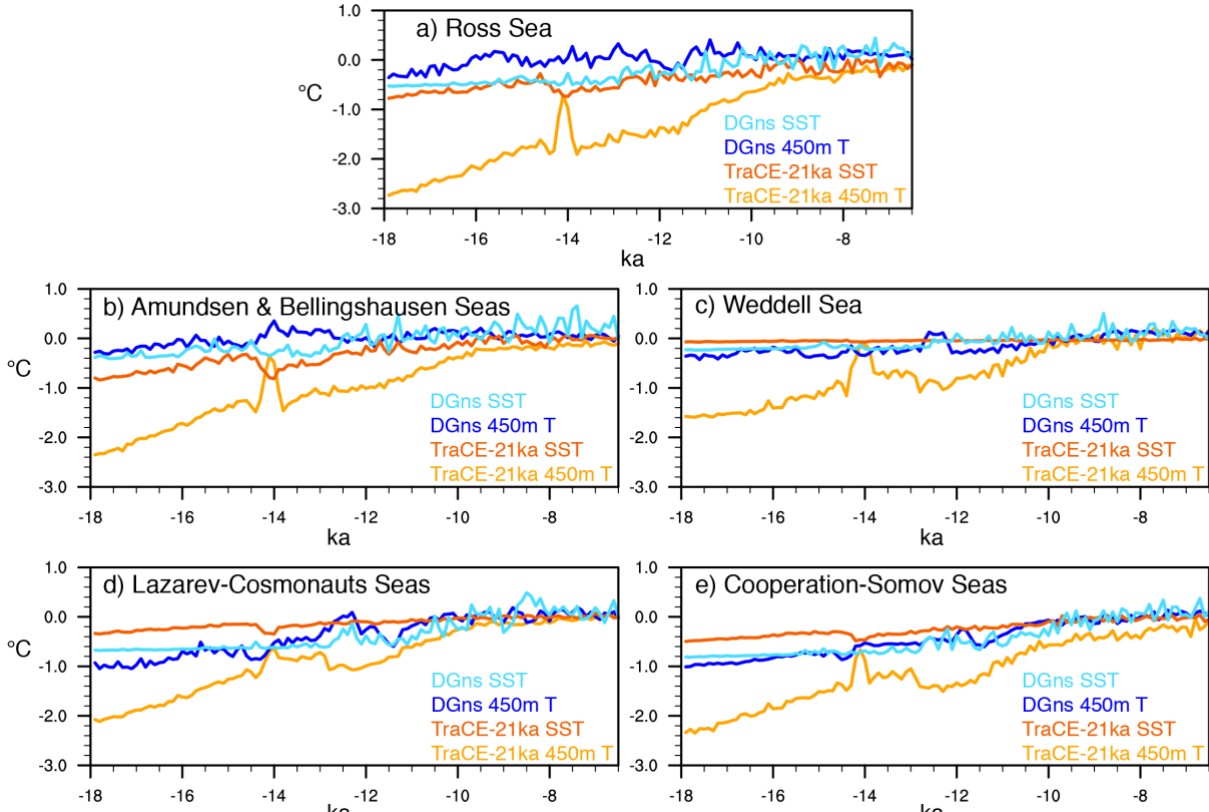

**Figure 7:** Time series of 100-yr mean annual average SST and 450 m depth ocean temperature anomalies relative to the Preindustrial era (°C) of the coastal seas around Antarctica, namely, (a) the Ross Sea (70°S—62°S, 168°E—160°W), (b) the Amundsen and Bellingshausen Seas (68°S—62°S, 135°W—60°W), (c) the

Weddell Sea (70°S—62°S, 60°W—30°W), (d) the coastal region from Lazarev Sea to Cosmonauts Seas (67°S—62°S, 15°W—50°E), and (e) the coastal region from Cooperation Sea to Somov Sea (67°S—62°S, 55°E—165°E).

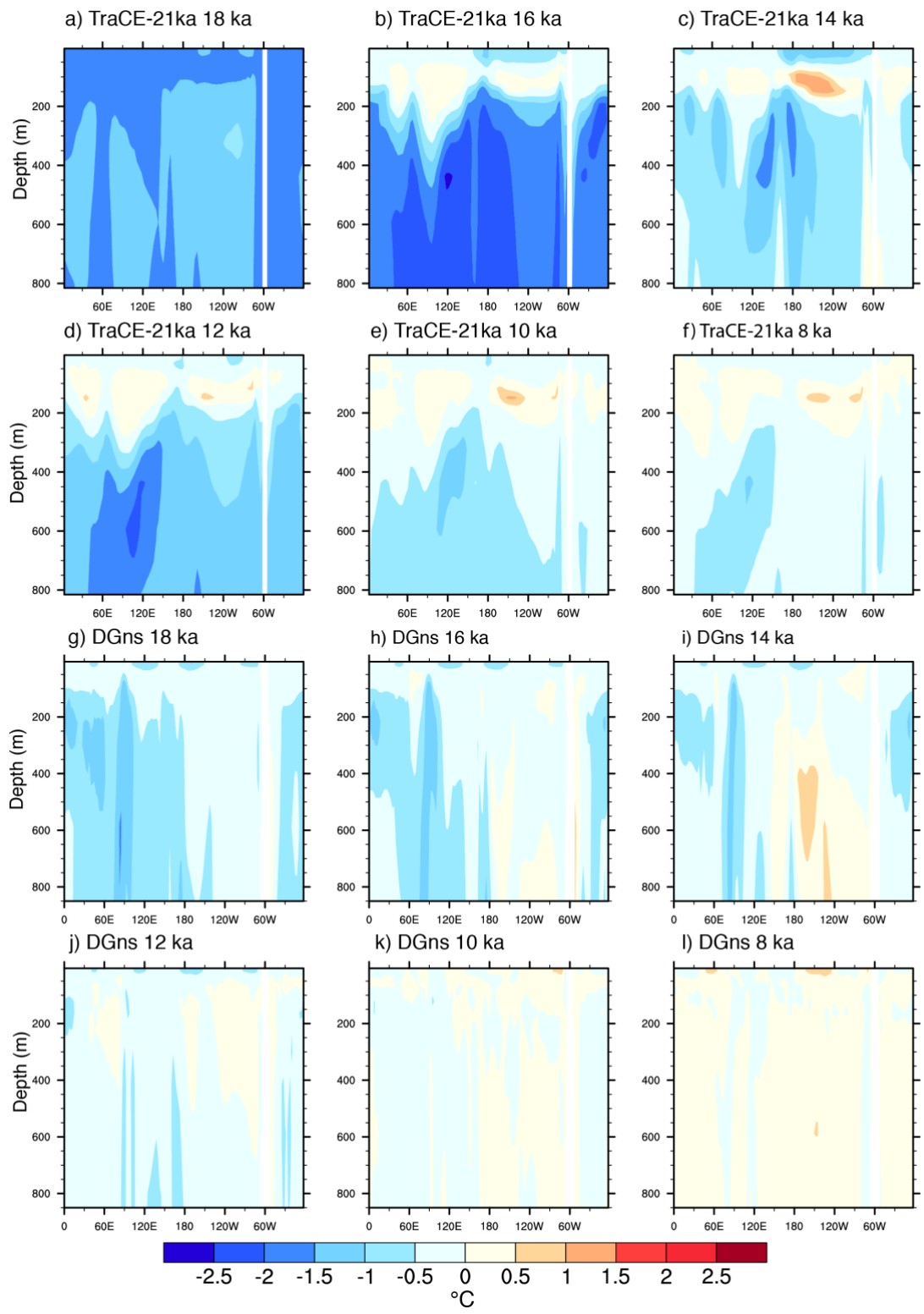


**Figure 8:** 2-ka time slices of longitudinal cross-sections of 100 yr-averaged ocean temperature anomalies relative to PI at 64°S the surface to 800 m depth: (a-f) TraCE-21ka 18ka to 8 ka; (g-l) DG_ns 18ka to 8ka.

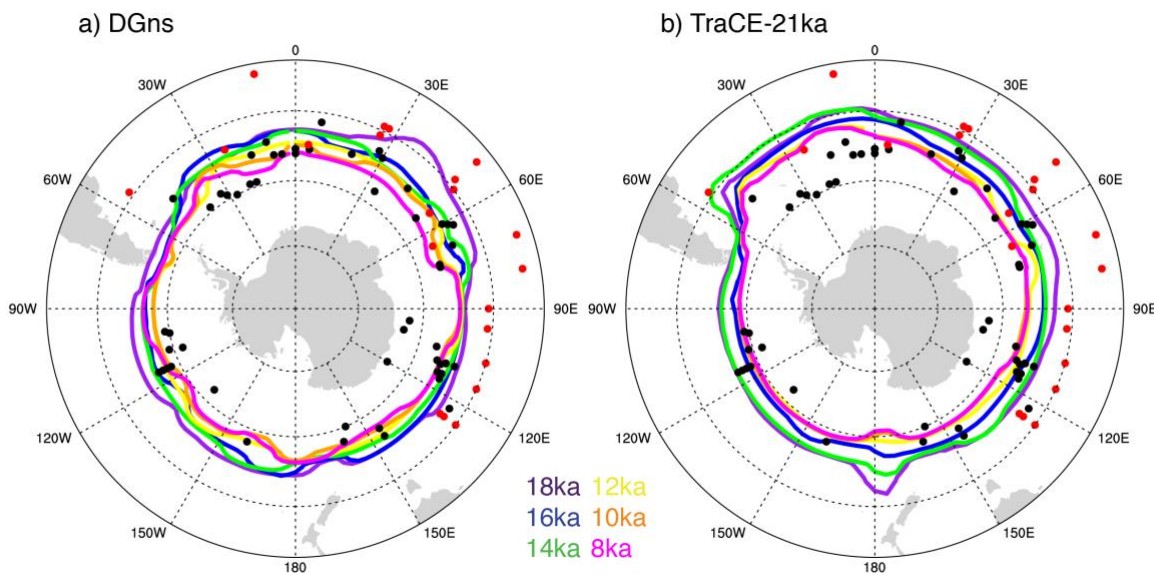

**Figure 9:** Modelled austral winter sea ice extent (15% coverage contour) per 2-ka for (a) DG$_{ns}$ and (b) TraCE-21ka from 18-8 ka (maximum Latitude of 35°S). Dotted lines indicate latitude (intervals of 15°) and longitude (intervals of 30°). The circles refer to marine sediment core sites with black circles corresponding to sites that indicate the presence of winter LGM sea ice, whereas the red circles represent sites that indicate no presence of winter LGM sea ice (adapted from Gersonde et al., 2005).

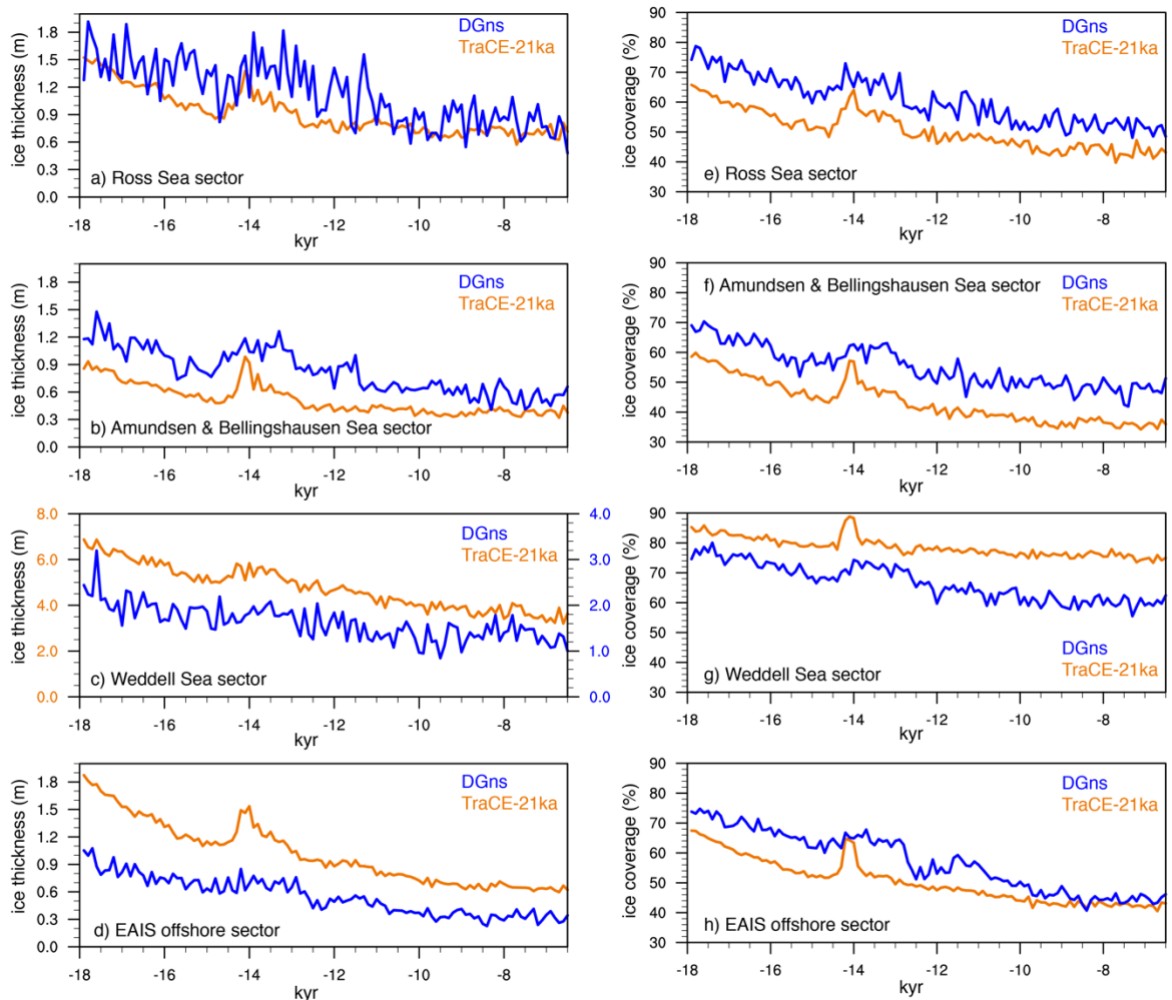

**Figure 10:** Time series of 100-yr mean annual average (a-d) sea ice thickness (m) and (e-h) coverage (%) in the Southern Ocean, namely, the Ross Sea sector (70°S—50°S, 168°E—160°W), the Amundsen and Bellingshausen Sea sector (68°S—50°S, 135°W—60°W), the Weddell Sea sector (70°S—50°S, 60°W—30°W), and the offshore EAIS sector from Lazarev Sea to Somov Sea (67°S—50°S, 15°W—165°E). Please note the difference in scale in panel c.


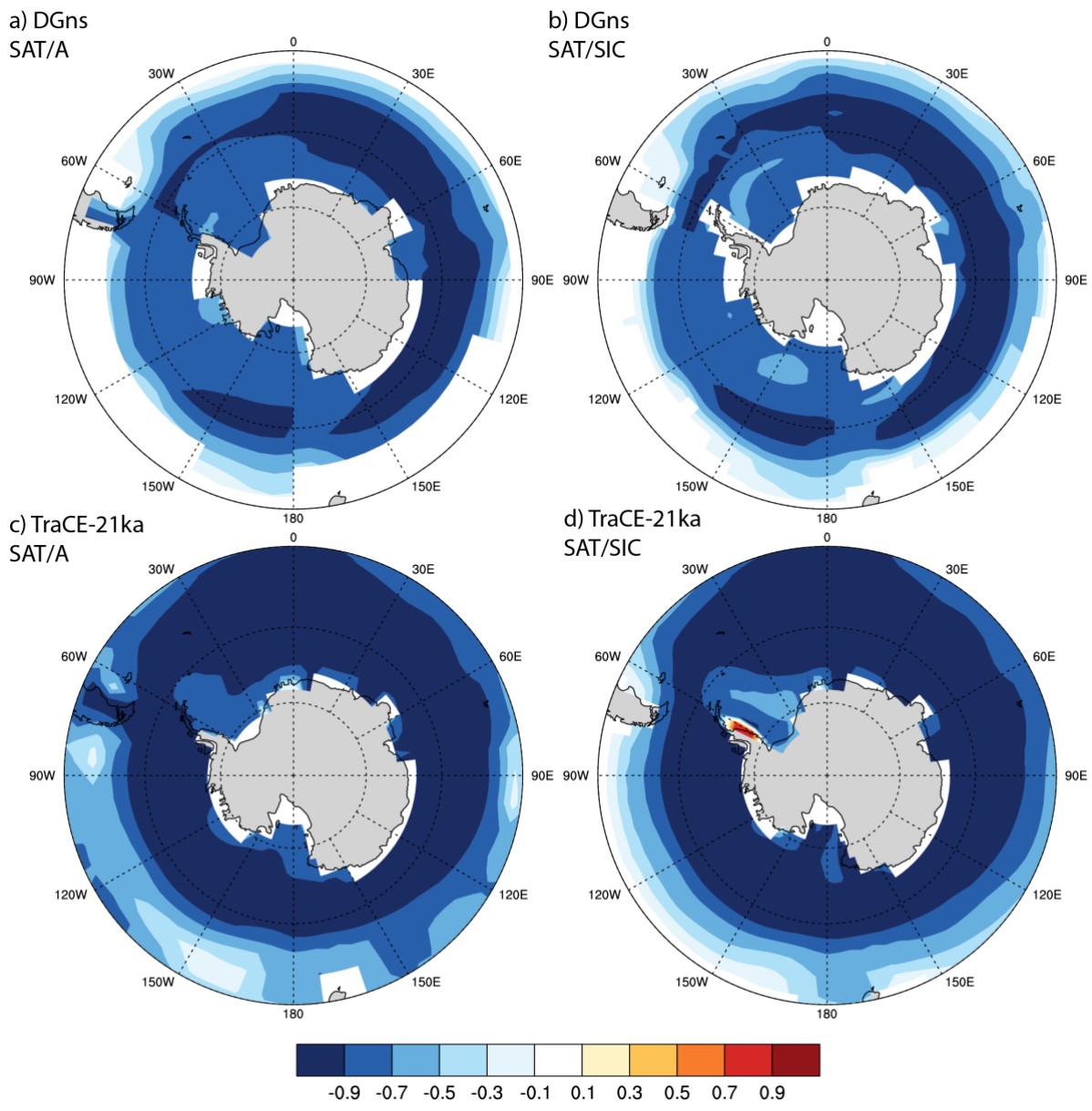

**Figure 11:** Spatial Pearson linear cross-correlation coefficients (r) between decadal surface air temperature (SAT, °C), surface albedo (A), and sea ice coverage (SIC, %) for (a-b) DG$_{ns}$ and (c-d) TraCE-21ka (maximum Latitude of 45°S). Dotted lines indicate latitude (intervals of 15°) and longitude (intervals of 30°). DG$_{ns}$ SAT was regridded to the same grid as DG$_{ns}$ SIC using bilinear interpolation in panel b.

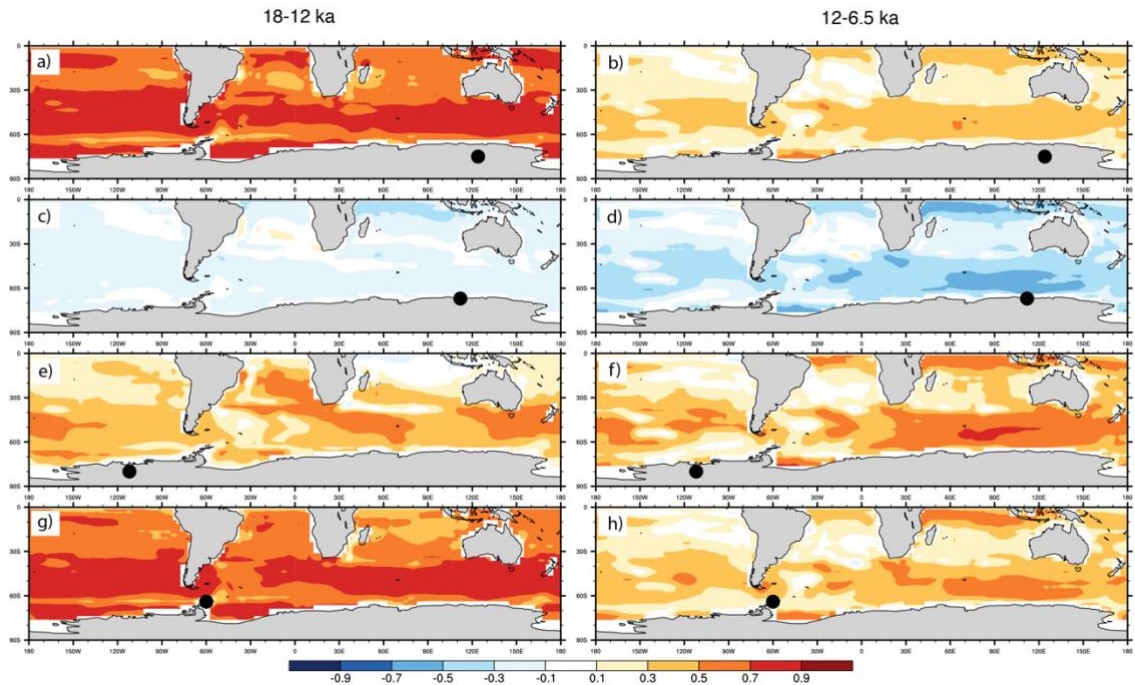

**Figure 12:** Spatial Pearson linear cross-correlation coefficients (r) between decadal SST and precipitation of the DG$_{ns}$ simulation for (left) 18-12 ka and (right) 12-6.5 ka at the (a,b) EDC (73-77°S, 121-127°E), (c,d) LD (65-70°S, 110-116°E), (e,f) WDC (77-82°S, 115-109°W), and (g,h) JRI (63-65°S, 59-62°W) ice core locations, respectively.

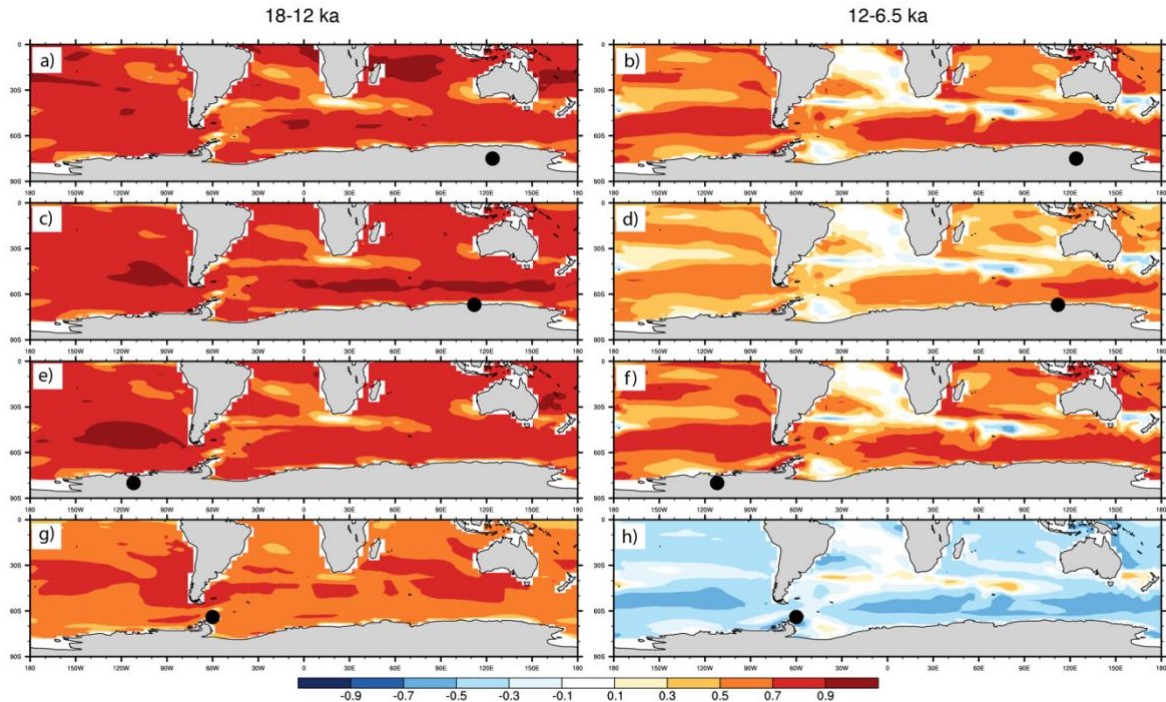

**Figure 13:** Spatial Pearson linear cross-correlation coefficients (r) between decadal SST and precipitation of the TraCE-21ka for (left) 18-12 ka and (right) 12-6.5 ka at the (a,b) EDC (73-77°S, 121-127°E), (c,d) LD (65-70°S, 110-116°E), (e,f) WDC (77-82°S, 115-109°W), and (g,h) JRI (63-65°S, 59-62°W) ice core locations, respectively.

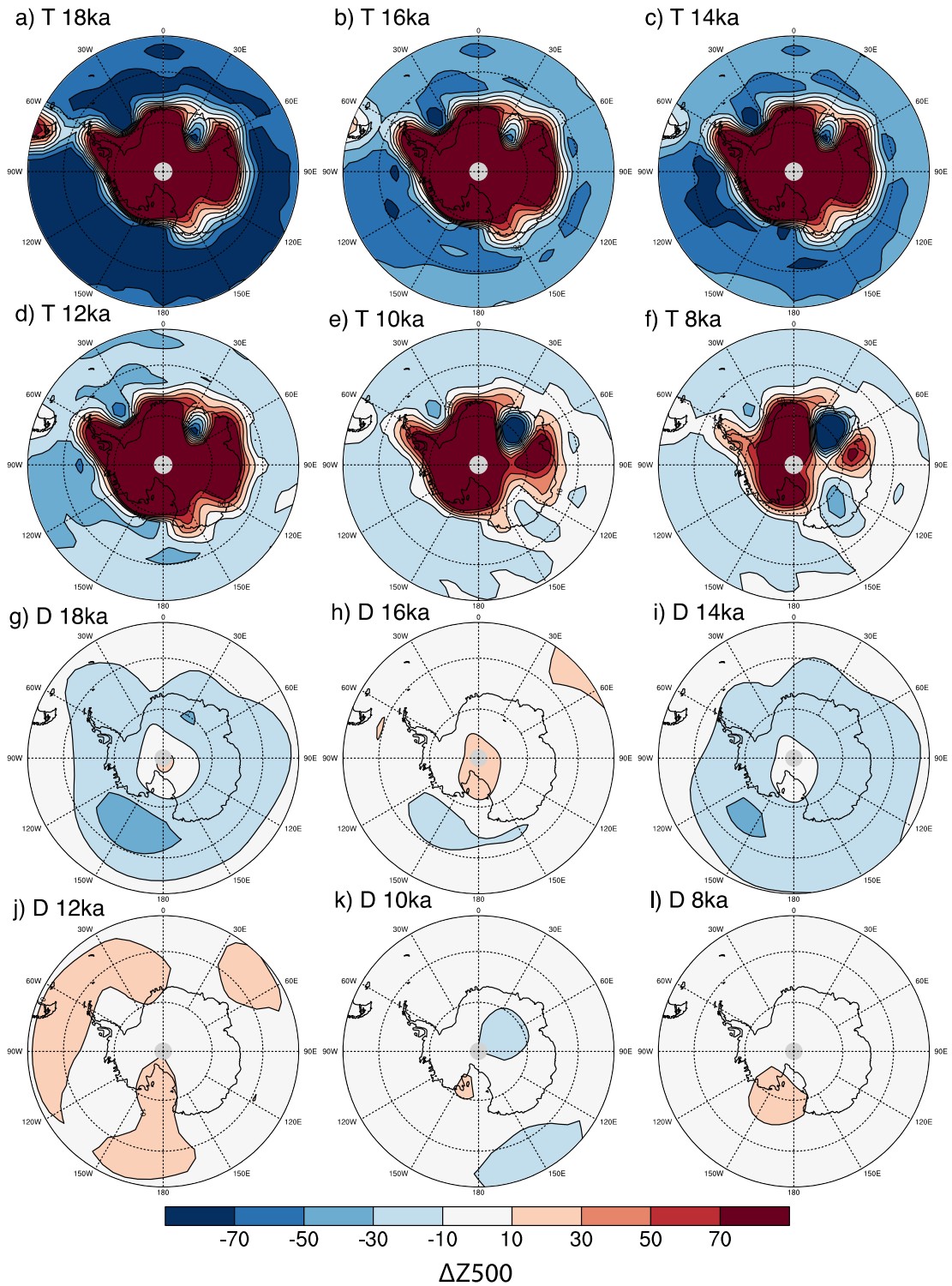

**Figure 14:** Geopotential height anomalies (m) at 500hPa relative to PI for the (a-f) TraCE-21ka and (g-l) DG$_{ns}$ deglacial experiments (maximum Latitude of 50°S). Dotted lines indicate latitude (intervals of 15°) and longitude (intervals of 30°).






