# Peer review of "Deglacial evolution of regional Antarctic climate and Southern Ocean conditions in transient climate simulations"

_Climate of the Past, 2018_

## Referee Comment (RC1) · Anonymous Referee #1 · 8 Aug 2018

This manuscript analyzed published transient simulations of the last deglaciation with a focus on regional conditions in Antarctic and Southern Ocean. The authors compared modeled temperature, accumulation rate and sea ice with available proxy estimates. Using model simulations, the authors also explored changes in variables and relationships that could impact ice-sheet mass balance.

The manuscript is well-written. The topic may interest readers of Climate of the Past. But, I hope the following questions and comments will be addressed.

Major comments:

1. In general, I feel the authors largely overlooked potential biases and uncertainty

in proxy records, including the ice-core temperature and Alkenone- and Mg/Ca-based SSTs. Stable isotopes in ice cores reflect complicated signals in climate system, such as changes in seasonality (Jouzel et al., 2003; Erb et al., 2018), sea-ice content and changes in moisture source regions (Noone and Simmonds, 2004; Holloway et al., 2016), etc. Similarly, marine SST records are also subject to substantial uncertainties (for example, see Tierney and Tingley, (2018) for a discussion of alkenone-based SSTs). The authors should better consider and incorporate these biases and uncertainties in their model-data comparison and related discussion. I suggest the authors further explore possible seasonality biases in ice-core and marine sediment records by comparing modeled seasonal temperatures, in addition to annual mean, with proxy records. They can also test whether water isotopes in ice cores more reflect temperature at condensation level or surface air temperature.

2. Related to the first comment, I suggest the authors provide more details on how ice-core $\delta$D is converted to temperature. What temporal temperature-$\delta$D slope is used for each proxy records? This could be done in Table 2.

3. The climate models used in the transient simulations were released more than 10 years ago (e.g., CCSM3 was released in 2004) and were considered outdated. I understand that there are no transient simulations using newer models, but some well-known biases in the models certainly deserves some caveats. For example, CCSM3 simulates much more sea-ice cover in both hemispheres than present-day observation (Yeager et al., 2006). Figure 7 of the authors' manuscript also shows a much more extensive sea-ice cover in the TraCE-21 LGM simulation than proxy estimates. Additionally, CCSM3 has problems simulating jet stream in the Southern Ocean and its response to external forcing (see Rojas et al., 2009). How are these model biases influence model-data comparison and findings in this manuscript?

4. For Figure 1a and b, I would like to see temperate changes at individual sites compared with model simulations. One way to do so is to use face color of markers to indicate proxy temperature changes and edge color (e.g., black or no edge) to represent whether model agrees with proxy estimates within uncertainty.

5. The authors are comparing averaged proxy SST in the Southern Ocean with model simulations in Figure 5a. I would suggest them also show model-data comparison at individual core sites, which enables us to see any potential regional difference in proxy estimates and possible divergent behavior from different proxy types (e.g., Leduc et al., 2010).

Minor comments:

1. Line 92: version T31x3 –> version 3

2. Line 99 and Table 1: What exact is the resolution of T21? 5.6° by 5.6°?

3. Line 162–163: Can you briefly justify the way you divide the Antarctic?

4. Figure1: it would be helpful if the authors can plot boxes/sectors for region EAIS interior, EAIS coastal, WAIS and AP.

5. Line 197: What is the assumed lapse rate of 1.0degC/100m based upon? I think this is too high. I suggest the authors to calculate the lapse rate in the model or reanalysis (e.g., Mokhov and Akperov, 2006).

6. Figure 5: Are these time series SST anomalies or absolute SST? If they are SSTA, how are they calculated?

7. Line 910–911: "within the range of proxy temperature reconstruction uncertainty of -10% to +30%" Where is the uncertainty range from? Jouzel et al. (2003)? Jouzel et al. (2003) estimated the uncertainty range for eastern Antarctic. How are the uncertainties for WAIS and AP obtained?

References:

Erb, M. P., Jackson, C. S., Broccoli, A. J., Lea, D. W., Valdes, P. J., Crucifix, M., & DiNezio, P. N. (2018). Model evidence for a seasonal bias in Antarctic ice cores. Nature

communications, 9(1), 1361.

Holloway, M. D., Sime, L. C., Singarayer, J. S., Tindall, J. C., Bunch, P., & Valdes, P. J. (2016). Antarctic last interglacial isotope peak in response to sea ice retreat not ice-sheet collapse. Nature communications, 7, 12293.

Jouzel, J., Vimeux, F., Caillon, N., Delaygue, G., Hoffmann, G., Masson-Delmotte, V., & Parrenin, F. (2003). Magnitude of isotope/temperature scaling for interpretation of central Antarctic ice cores. Journal of Geophysical Research: Atmospheres, 108(D12).

Leduc, G., Schneider, R., Kim, J. H., & Lohmann, G. (2010). Holocene and Eemian sea surface temperature trends as revealed by alkenone and Mg/Ca paleothermometry. Quaternary Science Reviews, 29(7), 989-1004.

Mokhov, I. I., & Akperov, M. G. (2006). Tropospheric lapse rate and its relation to surface temperature from reanalysis data. Izvestiya, Atmospheric and Oceanic Physics, 42(4), 430-438.

Noone, D., & Simmonds, I. (2004). Sea ice control of water isotope transport to Antarctica and implications for ice core interpretation. Journal of Geophysical Research: Atmospheres, 109(D7).

Rojas, M., Moreno, P., Kageyama, M., Crucifix, M., Hewitt, C., Abe-Ouchi, A., ... & Hope, P. (2009). The Southern Westerlies during the last glacial maximum in PMIP2 simulations. Climate Dynamics, 32(4), 525-548.

Yeager, S. G., Shields, C. A., Large, W. G., & Hack, J. J. (2006). The low-resolution CCSM3. Journal of Climate, 19(11), 2545-2566.
* * *

---

## Referee Comment (RC2) · Anonymous Referee #2 · 14 Sep 2018

Review of: Deglacial evolution of regional Antarctic climate and Southern Ocean conditions in transient climate simulations Daniel P. Lowry, Nicholas R. Golledge, Laurie Menviel, Nancy A.N. Bertler (doi: 10.5194/cp-2018-69)

Overview of manuscript: The authors analysed model output for the period 18 ka to 6.5 ka (they use "kyr"), which corresponds to the period from the last glacial termination (i.e., the short warming period that marks the transition to from the last ice age to the current inter-glacial period) to the mid-Holocene. Two models were used: TraCE-21ka (a fully coupled GCM) and LOVECLIM DGns (an intermediate complexity model). Analysis consisted of: (i) looking at time and space evolution of deglaciation in the

models, and (ii) comparing model output with proxies for surface temperature, surface mass balance, coastal ocean temperatures, and sea ice. The authors' were not able to draw firm conclusions about the mechanisms that determine the regional differences that paleoclimate records indicate existed for this period. They were also not able to determine the strengths and weaknesses of the models in terms of ice sheet mass balance predictions. This inability to draw firm conclusions was because there are few climate model simulations of this deglaciation period to make comparisons between, and because there is a lack of high-resolution proxy data.

General comments: After reading the abstract, I was very interested to hear what the authors' results were, but I ended up being extremely confused by the end of the manuscript and needed to re-read it several times. My confusion was mainly for the following reasons: (1) The aims and results outlined in the abstract do not appear to be consistent with what the conclusions state at the end of the manuscript; (2) Some of the figures and their captions are missing crucial information that makes them impossible to understand in isolation from the text; (3) There are some bold assertions regarding causation that do not appear to be supported by citations of the work of others or by independent analysis in this manuscript; (4) It is not clear to me how such sparse data sets can be compared to the models used. I have elaborated on these points in the specific comments below.

Specific comments: In this section, I provide specific details relating to the general comments above.

(1) The aims and results outlined in the abstract do not appear to be consistent with what the conclusions state at the end of the manuscript. The abstract states that the aim is to analyse results from two models to "better understand the mechanisms driving regional differences observed in paleoclimate models" and to "identify the main strengths and limitations of the models in terms of parameters that impact ice sheet mass balance". The abstract then states that the "climate simulations show" a number of results relating surface warming and accumulation rates to changes in sea ice, atmospheric circulation and ice surface elevation. The abstract also states that differences between the models and the proxy data exist, and suggested that this is because of inadequate representation of Meltwater Pulse 1A and 1B. However, in the "Summary and Conclusions" section, the ice sheet elevation effect on surface temperature is worded as if it is a specific result for TraCE-21ka, whereas in the abstract it is worded as if this is true for all simulations. In the "Summary and Conclusions" section, the accumulation rates are described as having "Strong discrepancies" between the models, which the authors suggest is related to model resolution issues, and they also note that the models do not match ice core accumulation reconstructions at the WDC and EDC sites. However, in the abstract the authors merely state that the accumulation changes in the model results are "quite distinct" and that the intermediate complexity model (which is not named in the abstract, but which is LOVECLIM DGns) had "resolution enhanced bias along the East Antarctic coast". The abstract states that variability in the relationship between accumulation and temperature has higher variability for coastal regions in the early to mid-Holocene, and state this "coincides with" atmospheric (Amundsen Sea Low) and sea ice changes. However, in the "Summary and Conclusions" section, this relationship is phrased more cautiously, with the use of "may", "appears to" and the statement for the need of a "more detailed moisture budget analysis". In the abstract, the mismatch between the models and proxies for the time and duration of the ACR and Younger Dryas/early Holocene warming is note, and states this is "suggesting that the Meltwater Pulse 1A and 1B events may be inadequately represented in these simulations." However, in the "Summary and Conclusions" section, the authors state that this mismatch "may result from model bias in large-scale ocean circulation, poorly constrained boundary conditions. . .or some combination of the two", and then mention meltwater forcing as something deglacial evolution is "highly sensitive to."

(2) Some of the figures and their captions are missing crucial information that makes them impossible to understand in isolation from the text: (i) The blue (DGns) and black (ice core data) lines are hard to distinguish in Figures 1-5. (ii) Figure 4 shows on the left-hand side graphs changes in snow accumulation (I think this should be "accumulation

rates" because the units are "%", so presumably "% per 100 years" as in figure 3?) on vertical axes and degrees Celcius temperature change on the horizontal axes, but these axes are not labelled (they should be). The top graph on the left (a) is missing minus signs from the lower part of that graph's vertical axis. Parts of graphs (f) and (h) (which are for EAIS coastal and AP, respectively) are shaded yellow, but it is not explained why in the caption. (iii) Figure 5 shows regional SST and ocean temperatures as a time series of 100 year averages (presumably means) for "TraCE" (called "TraCE-21ka" in previous graphs) and "LOVECLIM" (previous graphs called this DGns, the full title of the model is "LOVECLIM DGns"; consistency between graphs would be helpful). (iv) Figure 8 graphs are labelled (a) to (d) on both the left-hand side and the right-hand side, but the caption indicates that those on the left-hand side should be labelled (e) to (h), which is very confusing. The left-hand side graphs show 100 year averages of percentage sea ice coverage for (I presume, it does not say in the caption) the TraCE-21ka model and the DGns model, while those on the right-hand side (again, I presume) show sea ice thickness. The reader needs to assume the same color-coding for model output as in previous graphs, because there is no legend, which is confusing.

(3) There are some bold assertions regarding causation that do not appear to be supported by citations of the work of others or by independent analysis in this manuscript. This is particularly the case for causation attributed to sea ice changes. Examples include: (i) Lines 200-203: Large regional temperature differences in the model results for both models are stated to be "due to decreases in annual average sea ice coverage". How this conclusion regarding causation was reached is not explained. (ii) Lines 203-205: Differences between the results from the two models for regional temperature increases are stated to be "primarily due to differences in modelled sea ice". How this conclusion regarding causation was reached is not explained. Figure 8(c) indicates almost no change in Weddell Sea sea ice coverage for TraCE-21ka, but this is not discussed by the authors in this context. (iii) Having made some bold assertions regarding temperatures at lines 207-221, the authors then concede at lines 222-223 that "some of the differences between the models and ice core temperature reconstructions could be due to local climate effects of the ice core sites not captured in the broad regional averages of the climate models", which raises the question of how valid any of the comparisons between the ice cores and the models are. (iv) Lines 362-365: increases in continental surface temperature are linked with sea ice changes, with the authors stating "regions displaying the greatest increases in continental surface temperature that are not associated with changing ice sheet topography occur along the continental margins. . .suggesting that albedo-driven radiative changes associated with sea ice coverage may be an important driver of regional warming differences". This is more cautiously worded than the examples given in points (i) and (ii) above, but are still not physically justified. (v) Lines 369-370: similarly to point (iv) above, there is a lack of justification of the assertion "Changes in sea ice coverage may also explain the coastal warming differenced observed between DGns and TraCE-21ka." (vi) Lines 382-386: similarly to points (iv) and (v) above, there is a lack of physical justification for the assertion " the retreat of sea ice extent and reduced annual sea ice coverage in the early to mid-Holocene. . .may also introduce a greater variety of moisture sources of continental precipitation and alter the synoptic-scale variability, thereby weakening the SST-precipitation correlations in both models." (vii) Lines 454-471: in this paragraph, the authors start with "It may be expected that the retreat of sea ice and increased area of open ocean may introduce additional moisture sources, thereby enhancing precipitation relative to temperature." The authors then outline the main results from the literature, and summarize the results of their simulations which "do not exhibit a substantial increase in the scaling relationship with reduced sea ice coverage". In other words, the bold assertion of a conceptual model in their first sentence is not supported by their modelling results. The paragraph ends with a call for "additional moisture budget analysis".

(4) It is not clear to me how such sparse data sets can be compared to the models used. If I understand correctly what the authors have done, they have compared five ice cores with model output for surface temperatures from two global models, and two ice cores with model output for snow accumulation rates from two global models. As I have noted

earlier, the authors concede at lines 222-223 that "some of the differences between the models and ice core temperature reconstructions could be due to local climate effects of the ice core sites not captured in the broad regional averages of the climate models", which raises the question of how valid any of the comparisons between the ice cores and the models are. There is a great deal of research on comparing model results with observations for modern day climate, and I would particularly recommend the authors read Notz (2015), titled "How well must climate models agree with observations?" (doi: 10.1098/rsta.2014.0164) and the papers cited therein. Notz (2015) uses sea ice as a particular example, so it is very relevant for what the authors' are attempting to do here. Sea ice proxies particularly lacking, so what can the authors here really say?

―――――――――――――――

---

## Author Response (AR1)

Author's Response

Reviewer 1

We thank the Reviewer for their constructive comments regarding our manuscript. Our responses to specific comments are shown below in blue.

This manuscript analyzed published transient simulations of the last deglaciation with a focus on regional conditions in Antarctic and Southern Ocean. The authors compared modeled temperature, accumulation rate and sea ice with available proxy estimates. Using model simulations, the authors also explored changes in variables and relationships that could impact ice-sheet mass balance. The manuscript is well-written. The topic may interest readers of Climate of the Past. But, I hope the following questions and comments will be addressed.

Major comments:

1. In general, I feel the authors largely overlooked potential biases and uncertainty in proxy records, including the ice-core temperature and Alkenone- and Mg/Ca-based SSTs. Stable isotopes in ice cores reflect complicated signals in climate system, such as changes in seasonality (Jouzel et al., 2003; Erb et al., 2018), sea-ice content and changes in moisture source regions (Noone and Simmonds, 2004; Holloway et al., 2016), etc. Similarly, marine SST records are also subject to substantial uncertainties (for example, see Tierney and Tingley, (2018) for a discussion of alkenone-based SSTs). The authors should better consider and incorporate these biases and uncertainties in their model-data comparison and related discussion. I suggest the authors further explore possible seasonality biases in ice-core and marine sediment records by comparing modeled seasonal temperatures, in addition to annual mean, with proxy records. They can also test whether water isotopes in ice cores more reflect temperature at condensation level or surface air temperature.

We agree with the reviewer that a discussion of proxy uncertainties and inclusion of seasonal model output is warranted and will make our analysis more robust. We now discuss uncertainties related to ice core and marine SST records in Section 2.3 as well as in the context of our results and discussion. Specifically, we note that the temperature reconstructions of the ice cores can be influenced by the seasonality of precipitation, as explained in Jouzel et al. (2003) and discussed in a climate modelling context in Erb et al. (2018). As such, the ice cores could be reflecting a summer or winter temperature response, rather than an annual mean response. Considering that many of the available Southern Ocean SST reconstructions are alkenone-derived, we now explain that in regions of high seasonality, $TEX_{86}$ records can be biased low or high in accordance with the seasonal cycles of the marine archaea that produce GDGTs (Prahl et al., 2010). Seasonal output of both models (austral summer and winter) has been added in the time-series plots of Fig 1, 2 and 5. Additionally, we explain that changes in the temperatures and locations of the oceanic source region and sea ice concentration contribute to the uncertainty in the ice core temperature reconstructions (Uemura et al., 2012; Holloway et al., 2016). We also note to the Reviewer that this paper is primarily focused on the climate models, which are not isotope-enabled, and as such, analysis of the water isotopes of the ice cores is beyond the scope of this study.

The revised Fig 1 and Fig 5, which include the available seasonal output, are shown below. In consideration of the potential for seasonal bias in the proxy records, the climate models perform relatively well with respect to continental surface temperature. The temperature

change observed in the James Ross Island between 18 and 6.5 ka is within the range of seasonal temperature anomalies of both models (we previously stated that the models overestimate the temperature change at this site). The temperature changes at WAIS Divide, Epica Dome C, and Dome Fuji are also within the seasonal range of the DG$_{ns}$ simulation (we previously stated that the DG$_{ns}$ simulation underestimates the temperature change at these sites). Model performance with regard to SST is mixed, with the models generally showing less sea surface warming through the analysed period, with the exception of the highest latitude site in the Atlantic sector, where they overestimate the warming. The best model-proxy agreement is observed in the Pacific sector. Austral summer SSTs of the models generally show better agreement with the proxy reconstructions than the annual mean or austral winter SSTs. We also note that the SST proxy records are temporally sparse, which may also contribute to the model-data mismatch.

[Figure]

**Revised Figure 1 (now Fig 2):** Surface temperature change (°C) as simulated in (a) DG$_{ns}$ and (b) TraCE-21ka for the period of 18 to 6.5 ka. Ice core locations of the JRI, WDC, DF, V, and EDC sites are marked by filled circles, with the fill color corresponding to the change estimated in the ice core record, black outlines indicating a match in warming between the ice core and model simulation (i.e., the ice core temperature change from 18 to 6.5 ka is within the range of seasonal temperature changes of the climate model), and green (purple) outlines indicating an overestimation (underestimation) in warming by the models. For these model-proxy comparisons, we use site-specific model averages of land surface grid cells: JRI (63-65°S, 59-62°W), WDC (77-82°S, 115-

109°W), DF (75-79°S, 36-42°E), V (77-82°S, 104-110°E), EDC (73-77°S, 121-127°E). Stippling indicates a difference between decadal output of 18.0-17.5 ka and 7.0-6.5 ka that is significant at the 95% confidence level. (c-e) 100-yr average surface temperature time series (°C) of four regions (EAIS interior: 83°S-75°S, 30°W-165°E; coastal EAIS: 75°S-68°S, 15°W-165°E; WAIS: 83°S-72°S, 165°E-30°W; AP: 72°S-64°S, 64°W-59°W) relative to PI of both model simulations and ice core temperature reconstructions of sites located therein (DG$_{ns}$ in blue, TraCE-21ka in orange, ice cores in black). The colored shading indicates the seasonal range calculated from the 100-yr average austral summer and winter temperature anomalies. For EAIS, the ice core reconstruction is an average of the DF, V and EDC sites. The green box in (f) indicates the LGM temperature anomaly estimated for JRI (6.01±1.0°C; Mulvaney et al., 2012).

[Figure]

**Revised Figure 5 (now Fig. 6):** Sea surface temperature (SST) change (°C) as simulated in (a) DG$_{ns}$ and (b) TraCE-21ka for the period of 18 to 6.5 ka. Marine sediment locations are marked by open circles, with black outlines indicating a match in warming between the ice core and model simulation (i.e., the estimated proxy SST change from 18 to 6.5 ka is within the range of seasonal SST changes of the climate model), and green

(purple) outlines indicating an overestimation (underestimation) in warming by the models. Stippling indicates a difference between decadal output of 18.0-17.5 ka and 7.0-6.5 ka that is significant at the 95% confidence level. (c-h) Time series of 100-yr average mean annual SST from the models and SST proxy reconstructions (°C) at each individual marine sediment core site. The color shading represents the seasonal range calculated from the 100-yr average austral summer and winter temperature anomalies. In panel h, only a seasonal (February) proxy reconstruction is available; therefore, we only show modelled February SSTs from the climate models for this site.

2. Related to the first comment, I suggest the authors provide more details on how ice-core δD is converted to temperature. What temporal temperature-δD slope is used for each proxy records? This could be done in Table 2.

We have added the temporal isotope-temperature slopes to Table 2 for the ice core records. The listed citations in the Table also contain further details regarding the temperature and accumulation reconstructions. We rely on previously published records that are publicly available for download, hence these conversions are not performed as part of our analysis. The revised Table 2, which also includes 2 additional marine sediment core sites and one citation correction (TN057-6), is shown below:

Table 2. Antarctic and Southern Ocean proxy record details.

| Record name | Type | Measurement | Method | Location | Reference |
|---|---|---|---|---|---|
| Vostok (V) | Ice core | Temperature anomaly | δD temperature calibration (9.0‰ °C$^{-1}$) | East Antarctica, 78.5°S, 107°E | Lorius et al., 1995; Petit et al., 1999 |
| Dome Fuji (DF) | Ice core | Temperature anomaly | d temperature calibration (7.7‰ °C$^{-1}$) | East Antarctica, 77°S, 39°E | Uemura et al., 2012 |
| James Ross Island (JRI) | Ice core | Temperature anomaly | δD temperature calibration (6.7+/-1.3‰ °C$^{-1}$) | Antarctic Peninsula, 64°S, 58°W | Mulvaney et al., 2012 |
| EPICA Dome C (EDC) | Ice core | Temperature anomaly; Accumulation | δD temperature calibration (7.6‰ °C$^{-1}$); Ice flow model | East Antarctica, 75°S, 124°E | Jouzel et al., 2007; Parennin et al., 2007; Stenni et al., 2010 |
| WAIS Divide (WDC) | Ice core | Surface temperature; Accumulation | Water stable isotope record, borehole temperatures and nitrogen isotopes (7.9‰ °C$^{-1}$); Ice flow model | West Antarctica, 79.5°S, 112°W | WAIS Divide Project Members, 2013; Cuffey et al., 2016; Fudge et al., 2016 |
| Core MD03-2611 | Marine sediment core | SST | Alkenone-derived | Murray Canyons area, 36°S, 136°E | Calvo et al., 2007 |
| ODP site 1233 core | Marine sediment core | SST | Alkenone-derived | SE Pacific, 41°S, 74°W | Kaiser et al., 2005 |
| Core MD97-2120 | Marine sediment core | SST | Mg/Ca-derived | Chatham Rise, 45°S, 175°E | Pankhe et al., 2003 |
| Core TN057-6 | Marine sediment core | SST | Alkenone-derived | South Atlantic, 43°S, 9°E | Anderson et al., 2014 |
| Core MD07-3128 | Marine sediment core | SST | Alkenone-derived | Magellan Strait, 53°S, 75°W | Caniupan et al., 2011 |
| Core TN057-13 | Marine sediment core | Seasonal SST (Feb, Aug) | Diatom-derived | East Atlantic Polar Front, 50°S, 6°E | Nielson et al., 2004; Anderson et al., 2009 |

3. The climate models used in the transient simulations were released more than 10 years ago (e.g., CCSM3 was released in 2004) and were considered outdated. I understand that there are no transient simulations using newer models, but some well-known biases in the models certainly deserves some caveats. For example, CCSM3 simulates much more sea-ice cover in both hemispheres than present-day observation (Yeager et al., 2006). Figure 7 of the authors' manuscript also shows a much more extensive sea-ice cover in the TraCE-21 LGM simulation than proxy estimates. Additionally, CCSM3 has problems simulating jet stream in the Southern Ocean and its response to external forcing (see Rojas et al., 2009). How are these model biases influence model-data comparison and findings in this manuscript?

We agree with the Reviewer that our results need to be discussed in the context of these known model biases. While discussed the limitations of the intermediate complexity atmospheric component of LOVECLIM with regard to precipitation, we did not sufficiently address the known biases of CCSM3 or the LOVECLIM ocean component CLIO, as stated in Yeager et al. (2006), Danabasoglu et al., (2012), Dreisschaert et al. (2005) and Rojas et al. (2009). This has been added to Section 4.2 (see lines 559-573).

4. For Figure 1a and b, I would like to see temperate changes at individual sites compared with model simulations. One way to do so is to use face color of markers to indicate proxy temperature changes and edge color (e.g., black or no edge) to represent whether model agrees with proxy estimates within uncertainty.

We agree with the Reviewer that this will be useful for readers and have revised Fig 1a,b and 2a,b accordingly (see revised Fig 1 above). We now use the outline to represent the model-proxy agreement, but use the circle fill to show the change estimated in the proxy record.

5. The authors are comparing averaged proxy SST in the Southern Ocean with model simulations in Figure 5a. I would suggest them also show model-data comparison at individual core sites, which enables us to see any potential regional difference in proxy estimates and possible divergent behavior from different proxy types (e.g., Leduc et al., 2010).

We agree that comparing the model output to proxy records at individual sites will enhance this analysis. Based on this suggestion, we included a new figure to show seasonal and annual average SSTs of individual sites (see revised Fig 5 above). We also include two additional sites that were omitted in the original submission, namely, Core MD07-3128 (Caniupan et al., 2011) and Core TN057-13 (Anderson et al. 2009).

Minor comments:

1. Line 92: version T31x3 –> version 3

This has been changed.

2. Line 99 and Table 1: What exact is the resolution of T21? 5.6∘ by 5.6∘?

Yes, 5.6° x 5.6°. This has been added to Table 1.

3. Line 162–163: Can you briefly justify the way you divide the Antarctic?

We have added the following: Considering differences observed between Antarctic ice core records from East and West Antarctica and between coastal and interior regions, we focus on the following four regions in our analysis: the interior of the East Antarctic ice sheet (EAIS interior; 83°S-75°S, 30°W-165°E), coastal East Antarctica (coastal EAIS; 75°S-68°S, 15°W-165°E), West Antarctica (WAIS; 83°S-72°S, 165°E-30°W), and the Antarctic Peninsula (AP; 72°S-64°S, 64°W-59°W).

4. Figure1: it would be helpful if the authors can plot boxes/sectors for region EAIS interior, EAIS coastal, WAIS and AP.

In order to not distract from the color contours of surface temperature in Fig 1, we have added the following figure, which shows each region, ice core site, and marine sediment record site:

[Figure]

**Figure 1:** View of Antarctica and the Southern Ocean (maximum Latitude of 35°S). The colors indicate the land and ocean mask of the TraCE-21ka simulation (yellow and light blue, respectively). The continental regions, namely, the Antarctic Peninsula (AP), the West Antarctic Ice Sheet (WAIS), the East Antarctic Ice Sheet interior (EAIS-I), and the East Antarctic Ice Sheet coastal region (EAIS-C) are outlined in the colored boxes (blue, red, green, and purple, respectively). The locations of the Antarctic ice core and marine sediment records used in this analysis are indicated by the black dots.

5. Line 197: What is the assumed lapse rate of 1.0degC/100m based upon? I think this is too high. I suggest the authors to calculate the lapse rate in the model or reanalysis (e.g., Mokhov and Akperov, 2006).

We calculated the average polar lapse rate over Antarctica in LOVECLIM for the 18ka time-slice to be 0.54°C/100m, which is in closer agreement with the Mokhov and Akperov (2006) estimation. This sentence has been revised accordingly.

6. Figure 5: Are these time series SST anomalies or absolute SST? If they are SSTA, how are they calculated?

This figure has been divided into two figures, following the Reviewer's suggestions of examining individual sites. In the first, considering the discussion regarding seasonal biases of SST proxy reconstructions, we show the absolute SSTs. In the second figure, in which only the modelled coastal ocean temperatures are shown, we calculate SST anomalies relative to the Preindustrial Era.

7. Line 910–911: "within the range of proxy temperature reconstruction uncertainty of -10% to +30%" Where is the uncertainty range from? Jouzel et al. (2003)? Jouzel et al. (2003) estimated the uncertainty range for eastern Antarctic. How are the uncertainties for WAIS and AP obtained?

Cuffey et al. (2016) and Mulvaney et al. (2012) provide uncertainties for the temperature reconstructions for the WAIS Divide and James Ross Island ice cores, respectively. This is specified in the caption for the newly added Table 3.

**Table 3.** Change from 18 to 6.5 ka at Antarctic ice core sites (surface temperature, °C; accumulation, cm/yr) and marine sediment core sites (SST, °C) estimated in the proxy records and simulated in DG$_{ns}$ and TraCE-21ka for austral summer (December to February) and austral winter (June to August). Proxy records were linearly interpolated to 100 year averages. Bold font indicates a match between the seasonal range of change in the models and the proxy estimation at Antarctic ice core sites. Red font indicates that the seasonal range of change in the model does not overlap with the uncertainty range of the ice core record. We use uncertainties of +30% to -10% for V and DF temperature (Jouzel et al. 2003), +/- 2°C for EDC temperature (Stenni et al., 2010), +/-1.8°C for WDC temperature (Cuffey et al., 2016), and +/-1.0°C for JRI temperature (Mulvaney et al., 2012). For accumulation, we assume uncertainties of +/-25% (Fudge et al., 2016).

| Record | Proxy estimation (6.5 – 18 ka) | DG$_{ns}$ seasonal range (6.5 – 18 ka) | TraCE-21ka seasonal range (6.5 – 18 ka) |
|---|---|---|---|
| V | 8.04°C (7.24-10.45°C) | 2.75-6.25°C, 6.60-11.21 cm/yr | 9.02-10.40°C, 0.99-1.42 cm/yr |
| EDC | 8.18°C (6.18-10.18°C), 1.42 cm/yr (1.14-1.70 cm/yr) | **3.25-9.87°C**, 3.25-17.60 cm/yr | 8.99-10.12°C, 1.68-4.16 cm/yr |
| DF | 8.70°C (7.83 -11.31°C) | 3.70-8.23°C, 0.85-7.72 cm/yr | **8.33-10.69°C**, 3.58-4.09 cm/yr |
| WDC | 10.20°C (8.4-12.0°C), 11.8 cm/yr (8.85-14.8 cm/yr) | **8.40-11.68°C**, -5.00-8.11 cm/yr | 10.34-15.28°C, **11.37-19.91 cm/yr** |
| JRI | 6.26 (5.26-7.26°C) | 6.99-11.84°C, 10.07-21.25 cm/yr | **3.74-8.75°C**, 4.04-5.25 cm/yr |
| MD03-2611 | 7.77°C | 1.51-1.90°C | 2.26-3.05°C |
| ODP-1233 | 3.44°C | 2.90-4.02°C | 2.32-3.06°C |
| MD97-2120 | 3.40°C | 3.09-4.58°C | 1.64-2.52°C |
| TN057-6 | 5.69°C | 2.89-3.95°C | 2.03-2.44°C |
| MD07-3128 | 5.66°C | 4.68-5.08°C | 2.28-2.83°C |
| TN057-13 | 0.14°C | 2.97-3.49°C | 1.29-2.17°C |

References

Caniupán, M., Lamy, F., Lange, C. B., Kaiser, J., Arz, H., Kilian, R., ... & Laj, C. (2011). Millennial-scale sea surface temperature and Patagonian Ice Sheet changes off southernmost Chile (53 S) over the past~ 60 kyr. *Paleoceanography*, *26*(3).

Anderson, R. F., Ali, S., Bradtmiller, L. I., Nielsen, S. H. H., Fleisher, M. Q., Anderson, B. E., & Burckle, L. H. (2009). Wind-driven upwelling in the Southern Ocean and the deglacial rise in atmospheric $CO_2$. *science*, *323*(5920), 1443-1448.

Danabasoglu, G., Bates, S. C., Briegleb, B. P., Jayne, S. R., Jochum, M., Large, W. G., Peacock, S., Yeager, S. G. The CCSM4 ocean component. *Journal of Climate*, *25*(5), 1361-1389, 2012.

Driesschaert, E. Climate change over the next millennia using LOVECLIM, a new Earth system model including the polar ice sheets. Doctoral dissertation, Université Catholique de Louvain, Louvain-la-Neuve, Belgium, 2005.

Prahl, F. G., et al. "Systematic pattern in U37K′–temperature residuals for surface sediments from high latitude and other oceanographic settings." *Geochimica et Cosmochimica Acta*74.1 (2010): 131-143.

Uemura, R., Masson-Delmotte, V., Jouzel, J., Landais, A., Motoyama, H., & Stenni, B. (2012). Ranges of moisture-source temperature estimated from Antarctic ice cores stable isotope records over glacial–interglacial cycles. *Climate of the Past*, *8*(3), 1109-1125.

Mulvaney, R., Abram, N.J., Hindmarsh, R.C.A., Arrowsmith, C., Fleet, L., Triest, J., Sime, L.C., Alemany, O., Foord, S. Recent Antarctic Peninsula warming relative to Holocene climate and ice-shelf history. *Nature*, 489, 7414, 141-144, 2012. DOI: 10.1038/nature11391

Cuffey, K.M., G.D. Clow, E.J. Steig, C. Buizert, T.J. Fudge, M. Koutnik, E.D. Waddington, R.B. Alley, and J.P. Severinghaus. Deglacial temperature history of West Antarctica. *Proc. Natl. Acad. Sci.* 113(50): 14249-14254, 2016. doi:10.1073/pnas.1609132113.

Reviewer 2

We thank the Reviewer for their constructive comments regarding our manuscript. Our responses to specific comments are shown below in blue.

Overview of manuscript: The authors analysed model output for the period 18 ka to 6.5 ka (they use "kyr"), which corresponds to the period from the last glacial termination (i.e., the short warming period that marks the transition to from the last ice age to the current inter-glacial period) to the mid-Holocene. Two models were used: TraCE21ka (a fully coupled GCM) and LOVECLIM DGns (an intermediate complexity model). Analysis consisted of: (i) looking at time and space evolution of deglaciation in the models, and (ii) comparing model output with proxies for surface temperature, surface mass balance, coastal ocean temperatures, and sea ice. The authors' were not able to draw firm conclusions about the mechanisms that determine the regional differences that paleoclimate records indicate existed for this period. They were also not able to determine the strengths and weaknesses of the models in terms of ice sheet mass balance predictions. This inability to draw firm conclusions was because there are few climate model simulations of this deglaciation period to make comparisons between, and because there is a lack of high-resolution proxy data.

We hope that the Reviewer finds that the revised manuscript supports a number of firm conclusions regarding the mechanisms that determine regional differences in deglacial Antarctic climate and Southern Ocean changes as well as the strengths and limitations of the climate models. In particular, we demonstrate the sensitivity of coastal ocean temperatures to Southern Ocean meltwater forcing, regional differences in accumulation-temperature scaling relationships, and the strong correlation between surface air temperatures and surface albedo and sea ice coverage over the Southern Ocean. We also show that both models successfully capture the centennial-scale rates of temperature changes recorded in Antarctic ice core records, but also show key biases with regard to early Holocene SSTs and continental accumulation.

General comments: After reading the abstract, I was very interested to hear what the authors' results were, but I ended up being extremely confused by the end of the manuscript and needed to re-read it several times. My confusion was mainly for the following reasons: (1) The aims and results outlined in the abstract do not appear to be consistent with what the conclusions state at the end of the manuscript; (2) Some of the figures and their captions are missing crucial information that makes them impossible to understand in isolation from the text; (3) There are some bold assertions regarding causation that do not appear to be supported by citations of the work of others or by independent analysis in this manuscript; (4) It is not clear to me how such sparse data sets can be compared to the models used. I have elaborated on these points in the specific comments below. Specific comments: In this section, I provide specific details relating to the general comments above.

(1,2) We have edited the abstract and figures accordingly to avoid confusion. (3) We have revised the text in order to state our results more cautiously and now include additional figures to better illustrate relationships between climate parameters. (4) We agree with the Reviewer that the sparse spatial and temporal coverage of the paleoclimate and Southern Ocean marine proxy records is a challenge in assessing model performance, a caveat that we now discuss in greater detail in the revised manuscript. In addition to climate model-proxy comparison, this paper also highlights the similarities and differences between the two model simulations for a number of parameters relevant to Antarctic ice sheet mass balance for

which no proxy records are available, thereby addressing key data gaps in the observational record. Given that output of these climate model simulations continue to be applied as climate forcings in paleo-ice sheet simulations (e.g., Golledge et al., 2014; Tigchelaar et al., 2018; Petrini et al., 2018), this analysis will be useful for the community.

(1) The aims and results outlined in the abstract do not appear to be consistent with what the conclusions state at the end of the manuscript. The abstract states that the aim is to analyse results from two models to "better understand the mechanisms driving regional differences observed in paleoclimate models" and to "identify the main strengths and limitations of the models in terms of parameters that impact ice sheet mass balance". The abstract then states that the "climate simulations show" a number of results relating surface warming and accumulation rates to changes in sea ice, atmospheric circulation and ice surface elevation. The abstract also states that differences between the models and the proxy data exist, and suggested that this is because of inadequate representation of Meltwater Pulse 1A and 1B. However, in the "Summary and Conclusions" section, the ice sheet elevation effect on surface temperature is worded as if it is a specific result for TraCE-21ka, whereas in the abstract it is worded as if this is true for all simulations. In the "Summary and Conclusions" section, the accumulation rates are described as having "Strong discrepancies" between the models, which the authors suggest is related to model resolution issues, and they also note that the models do not match ice core accumulation reconstructions at the WDC and EDC sites. However, in the abstract the authors merely state that the accumulation changes in the model results are "quite distinct" and that the intermediate complexity model (which is not named in the abstract, but which is LOVECLIM DGns) had "resolution enhanced bias along the East Antarctic coast". The abstract states that variability in the relationship between accumulation and temperature has higher variability for coastal regions in the early to mid-Holocene, and state this "coincides with" atmospheric (Amundsen Sea Low) and sea ice changes. However, in the "Summary and Conclusions" section, this relationship is phrased more cautiously, with the use of "may", "appears to" and the statement for the need of a "more detailed moisture budget analysis". In the abstract, the mismatch between the models and proxies for the time and duration of the ACR and Younger Dryas/early Holocene warming is note, and states this is "suggesting that the Meltwater Pulse 1A and 1B events may be inadequately represented in these simulations." However, in the "Summary and Conclusions" section, the authors state that this mismatch "may result from model bias in large-scale ocean circulation, poorly constrained boundary conditions. . .or some combination of the two", and then mention meltwater forcing as something deglacial evolution is "highly sensitive to."

We have edited the abstract accordingly to avoid confusion. Specifically, removed the suggestion that "sea ice-albedo feedbacks likely drove regional surface temperature changes," but instead note that we observe strong correlations between surface air temperature and surface albedo over the Southern Ocean. We also clarify that the decrease in ice surface elevation only influenced surface temperature in one of the two climate models (as the DGns simulation has no evolving Antarctic ice mask, which is explained in the Methods section). Lastly, we have added that model bias in large-scale ocean circulation as well as biases in the observational records may also have contributed to the model-proxy mismatches.

(2) Some of the figures and their captions are missing crucial information that makes them impossible to understand in isolation from the text: (i) The blue (DGns) and black (ice core data) lines are hard to distinguish in Figures 1-5. (ii) Figure 4 shows on the lefthand side graphs changes in snow accumulation (I think this should be "accumulation rates" because

the units are "%", so presumably "% per 100 years" as in figure 3?) on vertical axes and degrees Celcius temperature change on the horizontal axes, but these axes are not labelled (they should be). The top graph on the left (a) is missing minus signs from the lower part of that graph's vertical axis. Parts of graphs (f) and (h) (which are for EAIS coastal and AP, respectively) are shaded yellow, but it is not explained why in the caption. (iii) Figure 5 shows regional SST and ocean temperatures as a time series of 100 year averages (presumably means) for "TraCE" (called "TraCE21ka" in previous graphs) and "LOVECLIM" (previous graphs called this DGns, the full title of the model is "LOVECLIM DGns"; consistency between graphs would be helpful). (iv) Figure 8 graphs are labelled (a) to (d) on both the left-hand side and the right-hand side, but the caption indicates that those on the left-hand side should be labelled (e) to (h), which is very confusing. The left-hand side graphs show 100 year averages of percentage sea ice coverage for (I presume, it does not say in the caption) the TraCE21ka model and the DGns model, while those on the right-hand side (again, I presume) show sea ice thickness. The reader needs to assume the same color-coding for model output as in previous graphs, because there is no legend, which is confusing.

We have edited the figure legends and captions accordingly to avoid confusion. (i) The blue lines have been brightened to better distinguish them from the black lines. (ii) In Figure 4, the units of accumulation are % relative to the Preindustrial era. This has been added to the Figure axes and caption. We also now describe the meaning of the yellow bars, which is meant to show the shift to higher variability in accumulation-temperature scaling. (iii/iv) The labelling in Fig 5 and 8 have been corrected, and a legend has been added to the latter. Revised versions of Figures 4, 5, and 8 are shown below. Given comments from Reviewer 1, we have divided Fig 5 into two figures to show the SSTs at each individual proxy site.

[Figure]

**Revised Figure 4 (now Fig 5):** (a-d) Scaling relationships of accumulation (% relative to PI) and temperature (°C relative to PI) in each region. Black and grey dots refer to the proxy record, blue and purple dots refer to the DG$_{ns}$ simulation, and orange and red dots refer to the TraCE21ka simulation. (e-h) The ratio of the change in precipitation (%) to the change in temperature (°C) per 500 years. The yellow bars in panel f and h indicate a shift to higher variability in accumulation-temperature scaling in the AP and EAIS coastal regions.

[Figure]

**Revised Figure 5-2 (now Fig 7):** Time series of 100-yr mean annual average SST and 450 m depth ocean temperature anomalies relative to the Preindustrial era (°C) of the coastal seas around Antarctica, namely, (a) the Ross Sea (70°S—62°S, 168°E—160°W), (b) the Amundsen and Bellingshausen Seas (68°S—62°S, 135°W—60°W), (c) the Weddell Sea (70°S—62°S, 60°W—30°W), (d) the coastal region from Lazarev Sea to Cosmonauts Seas (67°S—62°S, 15°W—50°E), and (e) the coastal region from Cooperation Sea to Somov Sea (67°S—62°S, 55°E—165°E).

[Figure]

**Revised Figure 8 (now Fig 10):** Time series of 100-yr mean annual average (a-d) sea ice thickness (m) and (e-h) coverage (%) in the Southern Ocean, namely, the Ross Sea sector (70°S—50°S, 168°E—160°W), the Amundsen and Bellingshausen Sea sector (68°S—50°S, 135°W—60°W), the Weddell Sea sector (70°S—50°S, 60°W—30°W), and the offshore EAIS sector from Lazarev Sea to Somov Sea (67°S—50°S, 15°W—165°E). Please note the difference in scale in panel c.

(3) There are some bold assertions regarding causation that do not appear to be supported by citations of the work of others or by independent analysis in this manuscript. This is particularly the case for causation attributed to sea ice changes. Examples include: (i) Lines 200-203: Large regional temperature differences in the model results for both models are stated to be "due to decreases in annual average sea ice coverage". How this conclusion regarding causation was reached is not explained. (ii) Lines 203-205: Differences between the results from the two models for regional temperature increases are stated to be "primarily due to differences in modelled sea ice". How this conclusion regarding causation was reached is not explained. Figure 8(c) indicates almost no change in Weddell Sea sea ice coverage for TraCE-21ka, but this is not discussed by the authors in this context. (iii) Having made some bold assertions regarding temperatures at lines 207-221, the authors then concede at lines 222-223 that "some of the differences between the models and ice core temperature reconstructions could be due to local climate effects of the ice core sites not captured in the broad regional averages of the climate models", which raises the question of how valid any of

the comparisons between the ice cores and the models are. (iv) Lines 362-365: increases in continental surface temperature are linked with sea ice changes, with the authors stating "regions displaying the greatest increases in continental surface temperature that are not associated with changing ice sheet topography occur along the continental margins. . .suggesting that albedo-driven radiative changes associated with sea ice coverage may be an important driver of regional warming differences". This is more cautiously worded than the examples given in points (i) and (ii) above, but are still not physically justified. (v) Lines 369-370: similarly to point (iv) above, there is a lack of justification of the assertion "Changes in sea ice coverage may also explain the coastal warming differenced observed between DGns and TraCE-21ka." (vi) Lines 382-386: similarly to points (iv) and (v) above, there is a lack of physical justification for the assertion " the retreat of sea ice extent and reduced annual sea ice coverage in the early to mid-Holocene. . .may also introduce a greater variety of moisture sources of continental precipitation and alter the synoptic-scale variability, thereby weakening the SST-precipitation correlations in both models." (vii) Lines 454-471: in this paragraph, the authors start with "It may be expected that the retreat of sea ice and increased area of open ocean may introduce additional moisture sources, thereby enhancing precipitation relative to temperature." The authors then outline the main results from the literature, and summarize the results of their simulations which "do not exhibit a substantial increase in the scaling relationship with reduced sea ice coverage". In other words, the bold assertion of a conceptual model in their first sentence is not supported by their modelling results. The paragraph ends with a call for "additional moisture budget analysis".

We have revised the text to advise more caution to our interpretations of the results. Specific responses are listed as follows:

(i) We agree with the Reviewer that we did not provide sufficient evidence to assert causality. As such, we have removed this statement. To better explore and illustrate the relationship between surface temperature and sea ice, we have added a figure to show the strong negative correlations that exist between surface air temperature (°C) and surface albedo and sea ice coverage (%) over the Southern Ocean in both models for the analysed period (see below figure).

[Figure]

**Figure 11:** Spatial Pearson linear cross-correlation coefficients (r) between decadal surface air temperature (SAT, °C), surface albedo (A), and sea ice coverage (SIC, %) for (a-b) DG$_{ns}$ and (c-d) TraCE-21ka. DG$_{ns}$ SAT was regridded to the same grid as DG$_{ns}$ SIC using bilinear interpolation in panel b.

(ii) We have removed this clause to avoid confusion. Figure 1 shows that in both models we observe more pronounced surface warming through the analysed period in the coastal regions surrounding Antarctica than in the continental interior. The exception is the region in TraCE-21ka that is impacted by ice mask changes. We acknowledge that many factors, including the changes in greenhouse gas content, orbital forcing and oceanic/atmospheric heat transport, contribute to surface air temperature changes in the coastal regions, but we now also demonstrate the strong statistical relationship between surface air temperature, surface albedo and sea ice coverage over the Southern Ocean (see Lines 412-430).

Regarding the Weddell Sea, we erroneously plotted the time-average sea ice fraction rather than the areal sea ice fraction of TraCE-21ka in Fig 8 in the original submission. This has been corrected, and we now show a more substantial decrease in sea ice coverage in this model (see revised Fig 8, now Fig 10). However, the sea ice coverage in both models shows

relatively lower correlations to surface air temperature in parts of the Weddell Sea, and in the case of TraCE-21ka, an area of positive correlation adjacent to the Antarctic Peninsula (i.e., warmer surface temperatures associated with higher sea ice coverage, and vice versa). In addition to the sea ice coverage, the surface albedo also depends on the state of the surface (e.g., snow depth, snow age, bare ice, melting, lead opening), which is now explained in the text (see Lines 426-431).

(iii) We have expanded on the caveats of model-proxy comparisons in both the Methods and Discussion sections (see Section 2.3, 4.2 and 4.3). We agree with the Reviewer that local climate effects may complicate comparisons to the broader regional averages in the climate models. However, this sentence is actually in reference to Fig 1c-f in the original submission, in which we plotted time series of regionally averaged modelled temperature anomalies rather than site-specific temperature anomalies. This sentence has been removed to avoid confusion.

We note that the Antarctic ice core locations were selected to be representative of global and regional climate, and with the exception of the James Ross Island ice core, the sites are located in areas that lack topographic features that would preclude model-proxy comparisons due to limitations in model resolution. Paleoclimate model simulations are often compared to ice core records as a means to evaluate climate model performance as well as to test the spatial representativeness of ice core locations (e.g., Masson-Delmotte et al., 2006). Both models used in this analysis have previously been applied to better understand regional temperature and accumulation changes in Antarctic ice core records, as discussed in sections 3.2, 3.3 and 4.1 (e.g., Goosse et al., 2012; Freiler et al., 2015; Fudge et al., 2016). Comparisons of this nature are an important benchmark for the Paleoclimate Model Intercomparison Project 3 (PMIP3; see Bracconot et al., 2012), a sub-set of the Coupled Model Intercomparison Project 5 (CMIP5), which is used to inform the Intergovernmental Panel on Climate Change (IPCC) in terms of future climate projections.

(iv) These lines have been revised in accordance the above Figure, which shows the strong negative correlations between surface temperature, surface albedo, and sea ice coverage.

(v) We have removed this sentence based on the Reviewer's comments.

(vi) In this sentence, we offer a suggestion that the retreat of sea ice is a possible contributor to explain the weakening of SST-precipitation correlations at each ice core location in the Holocene in addition to the lower millennial-scale variability relative to the early deglacial period. However, in consideration of the Reviewer's concerns regarding the lack of physical justification, we have removed the sentence.

(vii) In this paragraph, we are not proposing a new conceptual model but rather discussing an existing one, suggested by Monnin et al. (2004), and more recently in Palerme et al. (2017) in the context of future climate. Coastal Antarctic ice core records (e.g., Law Dome, Siple Dome, Taylor Dome) do exhibit enhanced precipitation-temperature scaling in the Holocene as compared to the Last Glacial Maximum. As the Reviewer states, we demonstrate here that this behaviour is not reproduced in these climate simulations, although we do observe an increase in the variability of the precipitation-temperature scaling relationship in the Holocene. Given the implications of the possible enhancement of precipitation-temperature scaling for the future sea level contribution of the Antarctic ice sheet, we suggest that this should be explored further; however, this is beyond the scope of the present study.

(4) It is not clear to me how such sparse data sets can be compared to the models used. If I understand correctly what the authors have done, they have compared five ice cores with model output for surface temperatures from two global models, and two ice cores with model output for snow accumulation rates from two global models. As I have noted earlier, the authors concede at lines 222-223 that "some of the differences between the models and ice core temperature reconstructions could be due to local climate effects of the ice core sites not captured in the broad regional averages of the climate models", which raises the question of how valid any of the comparisons between the ice cores and the models are. There is a great deal of research on comparing model results with observations for modern day climate, and I would particularly recommend the authors read Notz (2015), titled "How well must climate models agree with observations?" (doi: 10.1098/rsta.2014.0164) and the papers cited therein. Notz (2015) uses sea ice as a particular example, so it is very relevant for what the authors' are attempting to do here. Sea ice proxies particularly lacking, so what can the authors here really say?

We agree with the Reviewer that the Antarctic paleoclimate and Southern Ocean marine proxy records are sparse, both spatially and temporally, and this is actually the main motivation of this work, as we explain in the introduction. This is of course a challenge in all paleoclimate studies, and we plan to expand on these caveats in the Methods and Discussion sections. We specifically apply these climate simulations to address some of these gaps in the observational record and to better understand mechanisms.

It is also important to consider the time-scales of this analysis. Notz (2015) describes a number of factors to be considered in model-observation comparisons in the context of modern arctic sea ice, namely, climate model internal variability, tuning of a climate model for a particular purpose (e.g., matching modern arctic sea ice trends), observational uncertainty and uncertainty of forcings and boundary conditions. Given that these are multi-millennial simulations and the physical parameters of the models were not tuned to serve a particular purpose, only the uncertainties of the latter factors are relevant in this context. In terms of the observational uncertainty, we plan to expand on the caveats of SST reconstructions and ice core isotope records, including the potential for seasonal biases, as suggested by Reviewer 1. With regard to the forcings and boundary conditions, although the greenhouse gas forcing and solar insolation applied in the simulations are well-constrained, we explain in the Methods/Discussion sections how the uncertainties related to the ice sheet topography in both hemispheres and the timing, magnitude and location of freshwater forcing resulting from deglaciation are the most uncertain aspects (see Sections 2.1 and 4.3). As such, these boundary conditions are handled in different ways in the two simulations and lead to large differences between the models in terms of the analysed parameters.

We view the completeness of our data set and the use of two transient paleoclimate simulations, rather than one, as a main strength of our study. We offer a more comprehensive analysis of the climate parameters that impact Antarctic ice sheet mass than recent studies that have focused on a single aspect, proxy record or climate model. In addition to the assessment of climate model performance, our analysis of parameters for which no proxy records exist (e.g., coastal ocean temperatures at grounding line depths) is still highly valuable, as these models have been applied as forcings in numerous paleo-ice sheet modelling studies (e.g., Golledge et al., 2014; Tigchelaar et al., 2018; Petrini et al., 2018). As such, it is useful for the community to highlight the main differences between these two simulations, as they are consequential for ice sheet models. More specifically, we show here

that the differences in timing and amount of prescribed Southern Ocean meltwater forcing lead to differences in sub-surface ocean temperature anomalies, which is highly relevant for the marine-based West Antarctic Ice Sheet.

[revised manuscript text omitted]

---

## Referee Report (RR1)

Review of 'Daglacial Evolution of regional Antarctic climate and Southern Ocean conditions in transient climate simulations', by Lowry et al., Climate of the Past

This manuscript represents a useful contribution to our descriptive understanding of the extent to which transient paleo-simulations of Antarctic climate agree with paleoclimate proxy data for the surface of the Antarctic continent and the Southern Ocean. It provides quite a thorough comparison of two models and proxy data, showing where there is agreement and where there is not. The authors are not able to firmly conclude much about the driving mechanisms behind these discrepancies, apart from a clear impact of topography, and a likely feedback between sea ice and temperature. In general, I would have liked to see a little more discussion of ocean processes that could result in model bias.

Also, the authors mention correctly that subsurface ocean warmth can affect basal ice melt, and that subsurface temperature can evolve quite differently from SST. I would suggest that the authors look at each record of ocean temperature to check what depth it likely records (e.g. foram Mg/Ca ratios record the depth that the forams live, not necessarily SST), and check whether any new information can be gleaned in this way.

The figures and captions need some clarifying in order to be of real use to the reader. See suggestions and comments below. I would add to each map labels for latitude, and proxy record location labels for ease when referring to the figures from the text.

In my opinion, it would also help the paper if the authors could speculate more about what might constitute useful future steps, apart from simply running more simulations. For example, why stick to records of surface conditions? There are a few (very few) estimates of wind changes, which could be compared to the models in a similar way to sea ice extent. Also, there are growing numbers of deep-sea records which tell us about Southern Ocean circulation (e.g. Rae et al 2018, Nature and refs therein). While not appropriate for analysis/discussion in this study, the authors could round off this first significant attempt at Antarctic model-data comparison nicely by some more adventurous thinking about future steps.

**Minor points**

Fig 2: Caption – What do the authors consider a 'match', for putting black outlines round the filled points? Is this when the observations fit within the seasonal range? I think this becomes more clear in the text, but would be useful to state clearly here.

Figs 2, 3 and 6: Consider labelling cores on map. They are already labelled in Fig 1, but it would help not to have to skip back and forth while also referring to the text, especially when we get to Fig 6.

Figs 2 and 3, 6: What are the temperature changes documented in the time series panel relative to? I am not clear as to what is being compared here. Is it absolute temperature? If it is relative to the preindustrial era (as explained for Fig. 5 in response to a previous review comment), this needs stating in all relevant figure captions. Please clarify in the caption.

Fig 3: Does the coloured shading round the time series represent seasonal range, as for the temperature figures?

Line 98: Briefly outline what T21/T31 means (for non-modelers)

Line 105: Discuss the suitability of freshwater forcings based on McManus (there are more recent studies out there compiling a range of thorium/protactinium isotope data) and Greenland temperature. It would be useful to have some discussion (if brief) here, so those not in the know don't have to refer in detail back to the other papers. The following section describes how much meltwater is added to the model at different times, but doesn't go into why. Consider briefly adding this info.

Line 200: I would add the additional caveat to marine proxies that, because they are based on different species with differing depth habitats (which may change through time), they should be considered more carefully than simply being SST recorders. Have the authors thought in detail about this issue, or looked specifically at the records and species present to take depth into account?

Line 249: Can the authors comment on why the AP looks different to the other sites in this regard? i.e. DGns looks to have a large AP ACR signal in this case, despite having small change at other sites.

Line 260: The discussion of coastal precip anomalies based on model resolution seems ok, but what about the cause of the low precipitation at the pole?

Section 3.3: Please include many more references to specific figure panels throughout this discussion of scaling relationships, to help guide to reader.

Paragraph beginning Line 341: See point above regarding what the changes in temperature plotted in the time series are relative to. It is very difficult for the reader to see that four of the six sites show agreement to within 1 °C, when the records seem to be on different scales. Or are we looking at absolute temperatures for these figures?

Line 392: Rather than having the EAIS offshore sector in a sperate sentence, I would rather think it also is an exception.

Line 413: Should say Fig. 2 a,b

Section beginning line 559: I would add here some brief discussion of how coarse resolution climate models simulate the ACC and overturning, and changes in these when eddies are parameterised, versus changes when eddies are explicitly simulated. One important feature of model bias may be an overturning system that is too-sensitive to wind forcing (Farneti and Delworth 2010), which could feasibly lead to unrealistic changes in ocean temperature or fontal position. In my opinion, this is a source of likely very significant model bias, and should be discussed at least briefly in this paper. I would also note that many of the marine proxy records come from locations close to modern day major ocean frontal systems. Shifts in these systems across core locations would result in strong changes in SST/subsurface temperature, and so may lead to discrepancies between models and proxies, and to the regional differences in proxy records discussed. Given the data discussed here, it seems necessary that these ocean effects should be discussed.

---

## Author Response (AR2)

We thank the reviewer for their constructive comments. Our responses to specific comments are shown below in blue.

Review of 'Daglacial Evolution of regional Antarctic climate and Southern Ocean conditions in transient climate simulations', by Lowry et al., Climate of the Past

This manuscript represents a useful contribution to our descriptive understanding of the extent to which transient paleo-simulations of Antarctic climate agree with paleoclimate proxy data for the surface of the Antarctic continent and the Southern Ocean. It provides quite a thorough comparison of two models and proxy data, showing where there is agreement and where there is not. The authors are not able to firmly conclude much about the driving mechanisms behind these discrepancies, apart from a clear impact of topography, and a likely feedback between sea ice and temperature. In general, I would have liked to see a little more discussion of ocean processes that could result in model bias.

Also, the authors mention correctly that subsurface ocean warmth can affect basal ice melt, and that subsurface temperature can evolve quite differently from SST. I would suggest that the authors look at each record of ocean temperature to check what depth it likely records (e.g. foram Mg/Ca ratios record the depth that the forams live, not necessarily SST), and check whether any new information can be gleaned in this way.

We agree with the Reviewer that these are important points. A discussion of ocean processes that could result in model bias and the issue of forams recording temperature at different depths has been added to Sections 2.3 and 4.2. With regard to the latter point, the marine records used for model-data comparison of SST are published as SST reconstructions rather than temperature records at depth. But given the uncertainty, we have added the following caveats to Sections 2.3 and 4.2, respectively:

Lines 197-199: These SST reconstructions require cautious interpretations as they record temperatures at the depth at which the foramifera and diatoms live, and therefore may not offer straightforward reconstructions of SST.

Lines 570-572: The proxies may also be recording temperatures at depths other than the surface in accordance with the depth at which the diatoms and foraminfera occurred.

The figures and captions need some clarifying in order to be of real use to the reader. See suggestions and comments below. I would add to each map labels for latitude, and proxy record location labels for ease when referring to the figures from the text.

Done.

In my opinion, it would also help the paper if the authors could speculate more about what might constitute useful future steps, apart from simply running more simulations. For example, why stick to records of surface conditions? There are a few (very few) estimates of wind changes, which could be compared to the models in a similar way to sea ice extent. Also, there are growing numbers of deep-sea records which tell us about Southern Ocean circulation (e.g. Rae et al 2018, Nature and refs therein). While not appropriate for analysis/discussion in this study, the authors could round off this first significant attempt at Antarctic model-data comparison nicely by some more adventurous thinking about future steps.

We agree with the Reviewer that this would be useful and have included this discussion in Section 4.2 (see Lines 604-614).

Minor points Fig 2: Caption – What do the authors consider a 'match', for putting black outlines round the filled points? Is this when the observations fit within the seasonal range? I think this becomes more clear in the text, but would be useful to state clearly here.

Yes, this is included in the Fig 2 caption.

Figs 2, 3 and 6: Consider labelling cores on map. They are already labelled in Fig 1, but it would help not to have to skip back and forth while also referring to the text, especially when we get to Fig 6.

Done.

Figs 2 and 3, 6: What are the temperature changes documented in the time series panel relative to? I am not clear as to what is being compared here. Is it absolute temperature? If it is relative to the preindustrial era (as explained for Fig. 5 in response to a previous review comment), this needs stating in all relevant figure captions. Please clarify in the caption.

Figures 2c-f and 3c-f show surface temperature and accumulation relative to the Preindustrial Era, respectively, and Figure 5c-h show absolute SST, as indicated in the figure captions.

Fig 3: Does the coloured shading round the time series represent seasonal range, as for the temperature figures?

Yes. This is now clarified in the caption.

Line 98: Briefly outline what T21/T31 means (for non-modelers)

Done.

Line 105: Discuss the suitability of freshwater forcings based on McManus (there are more recent studies out there compiling a range of thorium/protactinium isotope data) and Greenland temperature. It would be useful to have some discussion (if brief) here, so those not in the know don't have to refer in detail back to the other papers. The following section describes how much meltwater is added to the model at different times, but doesn't go into why. Consider briefly adding this info.

We have added the following to Section 2.1: In both models, the freshwater is applied over large areas of the ocean surface in order to capture millennial-scale discharge events that occurred during the deglacial period, but the amounts and locations vary due to limited paleo-constraints.

Line 200: I would add the additional caveat to marine proxies that, because they are based on different species with differing depth habitats (which may change through time), they should be considered more carefully than simply being SST recorders. Have the authors thought in detail about this issue, or looked specifically at the records and species present to take depth into account?

We have added the following to Section 2.3 (Lines 197-199): These SST reconstructions require cautious interpretations as they record temperatures at the depth at which the foramifera and diatoms live, and therefore may not offer straightforward reconstructions of SST.

Line 249: Can the authors comment on why the AP looks different to the other sites in this regard? i.e. DGns looks to have a large AP ACR signal in this case, despite having small change at other sites.

We have added that the surface temperature in this region may be more heavily influenced by changes in sea ice associated with the ACR.

Line 260: The discussion of coastal precip anomalies based on model resolution seems ok, but what about the cause of the low precipitation at the pole?

This section has been revised as follows: The $DG_{ns}$ simulation shows a similar precipitation increase over the Southern Ocean (>8 cm/yr), however, precipitation decreases of 1-6 cm/yr occur over the South Pole and the coastal EAIS (Fig 3b). In the coastal region, where the decrease is larger, this is related to the coarse model resolution, which cannot adequately reproduce the steep slopes of East Antarctica and thus underestimates snow deposition in this region (Goosse et al., 2012). Changes in atmospheric circulation lead to the slightly reduced precipitation over the South Pole.

Section 3.3: Please include many more references to specific figure panels throughout this discussion of scaling relationships, to help guide to reader.

Done.

Paragraph beginning Line 341: See point above regarding what the changes in temperature plotted in the time series are relative to. It is very difficult for the reader to see that four of the six sites show agreement to within 1°C, when the records seem to be on different scales. Or are we looking at absolute temperatures for these figures?

This paragraph has been clarified to say that we are looking at the SST change between 18 and 6.5 ka.

Line 392: Rather than having the EAIS offshore sector in a separate sentence, I would rather think it also is an exception.

This sentence has been revised as follows: The exceptions are the Weddell Sea and along the EAIS coast, in which TraCE-21ka produces sea ice that is approximately double in thickness.

Line 413: Should say Fig. 2 a,b

This has been corrected.

Section beginning line 559: I would add here some brief discussion of how coarse resolution climate models simulate the ACC and overturning, and changes in these when eddies are parameterised, versus changes when eddies are explicitly simulated. One important feature of model bias may be an overturning system that is too-sensitive to wind forcing (Farneti and Delworth 2010), which could feasibly lead to unrealistic changes in ocean temperature or fontal position. In my opinion, this is a source of likely very significant model bias, and should be discussed at least briefly in this paper. I would also note that many of the marine proxy records come from locations close to modern day major ocean frontal systems. Shifts in these systems across core locations would result in strong changes in SST/subsurface temperature, and so may lead to discrepancies between models and proxies, and to the regional differences in proxy records discussed. Given the data discussed here, it seems necessary that these ocean effects should be discussed.

We agree with the Reviewer that this discussion is warranted. We have added the following to Section 4.2:

Lines 584-587: Lastly, given that the locations of the marine proxy records are in close proximity to modern ocean frontal systems, it should be noted that shifts in these systems through the deglacial period may account for some differences between the reconstructed and simulated ocean temperatures.

[revised manuscript text omitted]

a) DGns
SAT/A

b) DGns
SAT/SIC

c) TraCE-21ka
SAT/A

d) TraCE-21ka
SAT/SIC

-0.9  -0.7  -0.5  -0.3  -0.1  0.1  0.3  0.5  0.7  0.9

**Figure 11:** Spatial Pearson linear cross-correlation coefficients (r) between decadal surface air temperature (SAT, °C), surface albedo (A), and sea ice coverage (SIC, %) for (a-b) DG$_{ns}$ and (c-d) TraCE-21ka (maximum Latitude of 45°S). Dotted lines indicate latitude (intervals of 15°) and longitude (intervals of 30°). DG$_{ns}$ SAT was regridded to the same grid as DG$_{ns}$ SIC using bilinear interpolation in panel b.

[Figure]

**Figure 12:** Spatial Pearson linear cross-correlation coefficients (r) between decadal SST and precipitation of the DG$_{ns}$ simulation for (left) 18-12 ka and (right) 12-6.5 ka at the (a,b) EDC (73-77°S, 121-127°E), (c,d) LD (65-70°S, 110-116°E), (e,f) WDC (77-82°S, 115-109°W), and (g,h) JRI (63-65°S, 59-62°W) ice core locations, respectively.

[Figure]

**Figure 13:** Spatial Pearson linear cross-correlation coefficients (r) between decadal SST and precipitation of the TraCE-21ka for (left) 18-12 ka and (right) 12-6.5 ka at the (a,b) EDC (73-77°S, 121-127°E), (c,d) LD (65-70°S, 110-116°E), (e,f) WDC (77-82°S, 115-109°W), and (g,h) JRI (63-65°S, 59-62°W) ice core locations, respectively.

[Figure]

**Figure 14:** Geopotential height anomalies (m) at 500hPa relative to PI for the (a-f) TraCE-21ka and (g-l) DG$_{ns}$ deglacial experiments (maximum Latitude of 50°S). Dotted lines indicate latitude (intervals of 15°) and longitude (intervals of 30°).

**Page 23: [1] Deleted**      **Editor**      **1/14/19 3:58:00 PM**

Uemura, R., V. Masson-Delmotte, J. Jouzel, A. Landais, H. Motoyama, and B. Stenni. Ranges of moisture-source temperature estimated from Antarctic ice cores stable isotope records over glacial-interglacial cycles. *Climate of the Past*, 8, 1109-1125, 2012. Doi: 10.5194/cp-8-1109-2012